# A Normative and Biologically Plausible Algorithm for Independent Component Analysis

**Yanis Bahroun** [1,2]  **Dmitri B. Chklovskii** [1,4]  **Anirvan M. Sengupta** [2,3,5]

[1] Center for Computational Neuroscience, Flatiron Institute
[2] Center for Computational Mathematics, Flatiron Institute
[3] Center for Computational Quantum Physics, Flatiron Institute
[4] Neuroscience Institute, NYU Medical Center
[5] Department of Physics and Astronomy, Rutgers University
{ybahroun,dchklovskii}@flatironinstitute.org anirvans.physics@gmail.com

## Abstract

The brain effortlessly solves blind source separation (BSS) problems, but the algorithm it uses remains elusive. In signal processing, linear BSS problems are often solved by Independent Component Analysis (ICA). To serve as a model of a biological circuit, the ICA neural network (NN) must satisfy at least the following requirements: 1. The algorithm must operate in the online setting where data samples are streamed one at a time, and the NN computes the sources on the fly without storing any significant fraction of the data in memory. 2. The synaptic weight update is local, i.e., it depends only on the biophysical variables present in the vicinity of a synapse. Here, we propose a novel objective function for ICA from which we derive a biologically plausible NN, including both the neural architecture and the synaptic learning rules. Interestingly, our algorithm relies on modulating synaptic plasticity by the total activity of the output neurons. In the brain, this could be accomplished by neuromodulators, extracellular calcium, local field potential, or nitric oxide.

## 1 Introduction

In the brain, visual, auditory, and olfactory systems effortlessly identify latent sources from their mixtures [1, 2, 3, 4]. In unsupervised learning, such task is known as blind source separation (BSS) [5]. BSS is often solved by Independent Component Analysis (ICA) [6, 7], which assumes a generative model, wherein the observed stimuli are linear combinations of independent sources. ICA algorithms determine the linear transformation back from the observed stimuli into their original sources without knowing how they were mixed in the first place.

Developing a biologically plausible ICA algorithm may provide critical insight into neural computational primitives because ICA may be implemented throughout the brain. In particular, receptive fields of V1 neurons may be the result of performing ICA on natural images [8, 9]. Similarly, ICA may account for the receptive fields in the auditory system [10]. Moreover, the neural computational primitives used in the visual and the auditory cortex may be similar, as evidenced by anatomical similarity and by developmental experiments where auditory cortex neurons acquire V1-like receptive fields when visual inputs are redirected there [11]. Therefore, ICA may serve as a computational primitive underlying learning throughout the neocortex.

The majority of existing ICA algorithms[5], for example, information-theoretic ones [12, 13, 14, 15, 16] do not meet our biological plausibility requirements. In this work, for the biological plausibility of neural networks (NN), we require that i) they operate in the online (or streaming) setting, namely, the input dataset is streamed one data vector at a time, and the corresponding output must be computed

35th Conference on Neural Information Processing Systems (NeurIPS 2021).

without storing any significant fraction of the dataset in memory, ii) the weights of synapses in an NN must be updated using local learning rules, i.e., they depend only on the biophysical variables present in only the two neurons that the synapse connects or extracellular space near the synapse. Most existing bio-inspired ICA NNs [17, 18, 19, 20, 7] operate online but, when extracting multiple components, rely on non-local learning rules, i.e., a synapse needs to "know" about the individual activities of neurons other than the two it connects.

More biologically plausible ICA algorithms with local learning rules are limited to sources whose kurtosis deviates from the normal distribution in the same direction, are hand-crafted (ad hoc), and lack good theoretical guarantees[21, 22, 20]. An alternative to analyzing NNs with ad hoc learning rules is the normative approach. In the normative approach, an optimization problem with known offline solution is used as a starting point to derive online optimization algorithm which maps onto an NN with local learning rules. Such a normative approach led to the development of more biologically plausible ICA networks but only in limited settings of either nonnegative ICA [23, 24] or bounded component analysis [25, 26]

In this work, we develop a biologically plausible ICA neural network inspired by kurtosis-based ICA methods [27, 7, 28]. Specifically, we take inspiration in the Fourth-Order Blind Identification (FOBI) procedure which separates sources with distinct kurtosis [27]. In this context, distributions are often distinguished depending on their kurtosis relative to a Gaussian distribution, i.e., super- and sub-Gaussian distribution known respectively as leptokurtic ("spiky") and platykurtic ("flat-topped"). Our normative approach is based on a novel similarity-preserving objective for ICA with an intuitive geometric interpretation. We reformulate this objective as a min-max optimization problem and solve it online by stochastic gradient optimization. We demonstrate that our algorithm performs well on synthetic datasets, audio signals, and natural images.

Our online algorithm maps onto a single-layer NN that can separate independent sources without pre-processing. The synaptic weights in our NN are updated using local learning rules, extending more conventional Hebbian learning rules by a time-varying modulating factor, which is a function of the total output activity. The presence of such a modulating factor suggests a role of the extracellular environment on synaptic plasticity. Modulation of the plasticity rules by overall output activity agrees with several experimental studies that have reported that a third factor, in addition to pre- and post-synaptic activities, can play a crucial role in modulating the outcome of Hebbian plasticity. This could be accomplished by neuromodulators [29, 30, 31, 32], extracellular calcium [33], local field potential [34], or nitric oxide [35, 36, 37].

## 2   Problem statement and inspiration

The problem of BSS consists of recovering a set of unobservable source signals from observed mixtures. When mixing is linear, BSS can be solved by ICA, which decomposes observed random vectors into statistically independent variables.

Mathematically, ICA assumes the following generative model. There are $d$ sources recorded $T$ times forming the columns of $\mathbf{S} := [\mathbf{s}_1, \ldots, \mathbf{s}_T] \in \mathbb{R}^{d \times T}$ whose components $s_t^1, \ldots, s_t^d$ are assumed non-Gaussian and independent. Without loss of generality, we assume that each source has zero-mean, unit variance, and finite kurtosis. We also assume that sources have distinct kurtosis as is commonly done in kurtosis-based ICA methods [28]. The kurtosis of a random variable $v$ is defined as $\mathrm{kurt}[v] = \mathbb{E}\left[(v - \mathbb{E}(v))^4\right] / \left(\mathbb{E}\left[(v - \mathbb{E}(v))^2\right]\right)^2$. Finally, sources are linearly mixed, i.e., there exists a full rank mixing matrix, $\mathbf{A} \in \mathbb{R}^{d \times d}$, producing the $d$-dimensional mixture, $\mathbf{x}_t$:

$$\mathbf{x}_t = \mathbf{A}\mathbf{s}_t \qquad \forall t \in \{1, \ldots, T\} \ . \tag{1}$$

Then the goal of ICA algorithms is to determine a linear transformation of the observed signal, $\mathbf{W}_{ICA} \in \mathbb{R}^{d \times d}$, such that

$$\mathbf{y}_t := \mathbf{W}_{ICA}\mathbf{x}_t, \quad \forall t \in \{1, \ldots, T\} \ , \tag{2}$$

recovers unknown sources possibly up to a permutation and a sign flip.

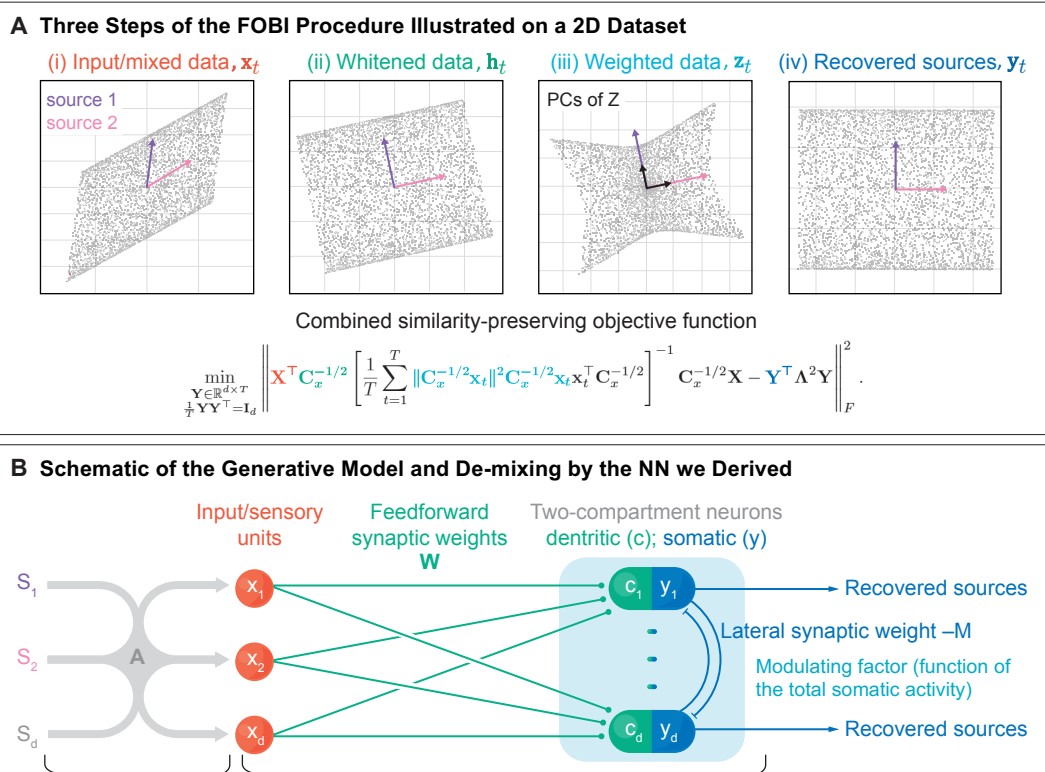

Figure 1: **A. The three steps of the FOBI procedure illustrated on a 2D dataset.** The purple and pink arrows show the axes of the independent sources $s_i$. **(i)** the observed signal, $\mathbf{x}_t$, **(ii)** the whitened data, $\mathbf{h}_t$, **(iii)** norm-weighted whitened data, $\mathbf{z}_t$, black arrows represent the principal directions of $\mathbf{z}_t$ **(iv)** the recovered sources, $\mathbf{y}_t$ are projections of $\mathbf{h}_t$ onto the principal directions of $\mathbf{z}_t$. These three steps are combined into a single objective function as indicated by color-coding **B. Schematic of the generative model and de-mixing by the NN we derived.** The output layer consists of two-compartment neurons whose synapses obey local learning rules. The dendritic compartments of the output neurons whiten data. The somatic compartments reconstruct the sources by rotating the whitened data. The pale blue rounded rectangle represents modulation of plasticity by output activity.

## 2.1 Review of the FOBI procedure

FOBI algorithm exploits a connection between ICA and Principal Component Analysis (PCA) [38, 39] pointed out in [27, 40]. We reproduce the proof of source recovery by FOBI from [27, 40] in Appendix A.

**Description of the procedure.** FOBI procedure consists of three steps, see **Fig. 1A**. First, the data must be whitened, i.e., all components become decorrelated and of unit variance, **Fig. 1A(ii)**. The whitening step can be performed using sample covariance, $\mathbf{C}_x = \frac{1}{T} \sum_{t=1}^{T} \mathbf{x}_t \mathbf{x}_t^\top$, as follows:

$$\textbf{Step 1: whiten} \quad \mathbf{h}_t = \mathbf{C}_x^{-1/2} \mathbf{x}_t \ .$$

Data whitening is often the first step in many ICA algorithms because recovering the sources after whitening corresponds to finding an orthogonal rotation matrix [17]. Various ICA methods differ in how the rotation is chosen. In FOBI, the whitened data, $\mathbf{h}_t$, is scaled by its norm and termed $\mathbf{z}_t$. Then, the directions of $\mathbf{h}_t$ and $\mathbf{s}_t$ with distinct kurtosis are recovered by finding the eigenvectors of

the sample covariance matrix $\frac{1}{T}\sum_{t=1}^{T}\mathbf{z}_t\mathbf{z}_t^\top$, **Fig. 1A(iii)**.

**Step 2a:** transform $\mathbf{z}_t = \|\mathbf{h}_t\| \cdot \mathbf{h}_t$ ; **Step 2b:** optimize $\mathbf{W}_z := \underset{\substack{\mathbf{W}\in\mathbb{R}^{d\times d} \\ \mathbf{W}\mathbf{W}^\top=\mathbf{I}_d}}{\arg\max}\; \frac{1}{T}\text{Tr}\left(\mathbf{W}\sum_{t=1}^{T}\mathbf{z}_t\mathbf{z}_t^\top\mathbf{W}^\top\right)$ .

Finally, to recover the sources, we project the whitened data, $\mathbf{h}_t$, onto the rows of $\mathbf{W}_z$, **Fig. 1A(iv)**

$$\textbf{Step 3: project} \quad \mathbf{y}_t = \mathbf{W}_z\mathbf{h}_t \ .$$

**Can a biologically plausible NN implement the FOBI algorithm?** The first two steps of FOBI do not present a problem with a biological implementation. For example, **Step 2b**, essentially a PCA, can be solved by a stochastic gradient ascent algorithm using Oja's learning rule [41]:

$$\Delta\mathbf{W}_z = \eta\left(\mathbf{u}_t\mathbf{z}_t^\top - \mathbf{u}_t\mathbf{u}_t^\top\mathbf{W}_z\right) \ . \tag{3}$$

Then, **Step 2b** can be mapped onto a single-layer network with upstream neurons' activity encoding $\mathbf{z}_t$, and the output neurons computing the components of $\mathbf{u}_t := \mathbf{W}_z\mathbf{z}_t$ where the elements of $\mathbf{W}_z$ are encoded in the weights of feedforward synapses. Eq.(3) gives the weight update for the feedforward synapses with the learning rate, $\eta > 0$. However, according to **step 3**, the final output of FOBI must be $\mathbf{y}_t$, obtained by multiplying the whitened inputs, $\mathbf{h}_t$, by $\mathbf{W}_z$ without scaling them by their norm. Such output may be computed by another single-layer network with the same feedforward synaptic weights, $\mathbf{W}_z$, but that would require weight-sharing (or weight transport). Alternatively, avoiding weight transport would require a non-local update rule for $\mathbf{W}_z$. Thus, both alternatives lead to biologically implausible solutions.

## 2.2 Similarity matching for principal subspace analysis

To find a biologically plausible implementation of FOBI, we follow the approach used previously to derive biologically plausible networks for Principal Subspace Projection (PSP) [42], a variant of PCA, and other tasks[43, 44, 45, 46, 47, 48, 49, 50, 51, 52, 53, 54]. This approach starts from reformulating the optimization objective for PSP in the so-called Similarity Matching (SM) form [55]:

$$\min_{\mathbf{Y}\in\mathbb{R}^{m\times T}}\left\|\mathbf{X}^\top\mathbf{X} - \mathbf{Y}^\top\mathbf{Y}\right\|_F^2 \ , \tag{4}$$

where $\mathbf{X} := [\mathbf{x}_1,\ldots,\mathbf{x}_T]$ is the data matrix, $\mathbf{Y} := [\mathbf{y}_1,\ldots,\mathbf{y}_T]$ is the output matrix, and $\|\cdot\|_F$ the Frobenius norm. In turn, the objective (4) can be optimized by an online algorithm that maps onto a single-layer network of linear neurons whose synapses obey local learning rules [44].

Whereas the SM approach leads to biologically plausible NNs for solving eigenproblems, it was still unclear how to overcome the weight transport challenge arising in the FOBI algorithm implementation. In the next Section, we address this challenge by introducing a novel SM objective for ICA.

## 3 A similarity-preserving objective for ICA

To derive a single-layer NN for ICA, which can be trained with local learning rules, we adopt a normative approach. We design a novel objective function, the solution of which projects the whitened data, $\mathbf{h}_t$, onto the eigenvectors of the covariance of $\mathbf{z}_t$ as specified in the FOBI procedure above using the SM approach. Specifically, we propose the following generalized nonlinearly weighted similarity matching objective using the notations, $\mathbf{H} := [\mathbf{h}_1,\ldots,\mathbf{h}_T]$ and $\mathbf{Z} := [\mathbf{z}_1,\ldots,\mathbf{z}_T]$ as defined earlier in Sec. 2.1 and illustrated in **Fig. 1A**,

$$\min_{\mathbf{Y}\in\mathbb{R}^{d\times T}}\left\|\mathbf{H}^\top\left[\frac{1}{T}\mathbf{Z}\mathbf{Z}^\top\right]^{-1}\mathbf{H} - \mathbf{Y}^\top\mathbf{\Lambda}^2\mathbf{Y}\right\|_F^2 \ , \quad \text{s.t.} \quad \frac{1}{T}\mathbf{Y}\mathbf{Y}^\top = \mathbf{I}_d \ , \tag{5}$$

with $\mathbf{\Lambda}^2 = \text{diag}(\lambda_1^2,\ldots,\lambda_d^2)$ any diagonal matrix with distinct finite positive entries.

To accomplish all the three steps above in a single-layer network, we rewrite (5) in terms of the input data $\mathbf{X}$ by substituting the expressions for $\mathbf{H}$ (**Step 1**) and $\mathbf{Z}$ (**Step 2a**):

$$\min_{\substack{\mathbf{Y}\in\mathbb{R}^{d\times T} \\ \frac{1}{T}\mathbf{Y}\mathbf{Y}^\top=\mathbf{I}_d}}\left\|\mathbf{X}^\top\mathbf{C}_x^{-1/2}\left[\frac{1}{T}\sum_{t=1}^{T}\|\mathbf{C}_x^{-1/2}\mathbf{x}_t\|^2\mathbf{C}_x^{-1/2}\mathbf{x}_t\mathbf{x}_t^\top\mathbf{C}_x^{-1/2}\right]^{-1}\mathbf{C}_x^{-1/2}\mathbf{X} - \mathbf{Y}^\top\mathbf{\Lambda}^2\mathbf{Y}\right\|_F^2 \ . \tag{6}$$

Then the global minima of objective (6) recover the sources as formalized by the following theorem:

**Theorem 1.** *Given that the sources are independent, centered, have unit variance, and distinct kurtosis (c.f. Sec 2), then the global optimal solution for our objective (6), denoted by $\mathbf{Y}^*$ satisfies*

$$\mathbf{Y}^* = \mathbf{\Xi}\mathbf{\Pi}\mathbf{S} \tag{7}$$

*where $\mathbf{\Xi}$ is a diagonal matrix with $\pm 1$'s on the diagonal, and $\mathbf{\Pi}$ is a permutation matrix, and is thus a solution to the ICA problem.*

*Proof.* We give the detailed proof of the theorem in Appendix A. $\qquad\square$

## 4 Derivation of the algorithm

While our objective (6) can be minimized by taking gradient descent steps with respect to $\mathbf{Y}$, this would not lead to an online algorithm because such computation requires combining data from different time steps. Instead, following [44], we introduce auxiliary matrix variables corresponding to synaptic weights, which store sufficient statistics allowing for the ICA computation using solely instantaneous inputs. Such substitution leads to a min-max optimization problem that is solved by gradient descent/ascent. A corresponding online optimization algorithm using stochastic gradient descent/ascent maps onto an NN with local learning rules.

### 4.1 Min-max formulation

Here we modify the objective (6) by introducing auxiliary variables, namely $\mathbf{W}$ and $\mathbf{M}$, leading to a min-max optimization problem. In the following sub-sections, the gradient descent/ascent optimization of the min-max objective will lead to an online algorithm that maps onto an NN where $\mathbf{W}$ and $\mathbf{M}$ correspond to synaptic weights.

We expand the square in Eq. (6), normalizing by $T^2$, and dropping terms that do not depend on $\mathbf{Y}$ yielding:

$$\min_{\mathbf{Y}\in\mathbb{R}^{d\times T}} \frac{1}{T^2}\operatorname{Tr}\left(-2\mathbf{X}^\top\mathbf{\Gamma}_x\mathbf{X}\mathbf{Y}^\top\mathbf{\Lambda}^2\mathbf{Y} + \mathbf{Y}^\top\mathbf{\Lambda}^2\mathbf{Y}\mathbf{Y}^\top\mathbf{\Lambda}^2\mathbf{Y}\right) \quad \text{s.t.} \quad \frac{1}{T}\mathbf{Y}\mathbf{Y}^\top = \mathbf{I}_d \ , \tag{8}$$

where, for convenience, we introduce

$$\mathbf{\Gamma}_x := \mathbf{C}_x^{-1/2}\left[\frac{1}{T}\sum_{t=1}^{T}\|\mathbf{C}_x^{-1/2}\mathbf{x}_t\|^2\mathbf{C}_x^{-1/2}\mathbf{x}_t\mathbf{x}_t^\top\mathbf{C}_x^{-1/2}\right]^{-1}\mathbf{C}_x^{-1/2} \ .$$

The quartic term in $\mathbf{Y}$ in (8) is a constant under the decorrelation constraint and can be dropped from the optimization.

We now introduce auxiliary matrix variables $\mathbf{W}$ and $\mathbf{M}$, resulting in:

$$\min_{\mathbf{Y}\in\mathbb{R}^{d\times T}} \min_{\mathbf{W}\in\mathbb{R}^{d\times d}} \max_{\mathbf{M}\in\mathbb{R}^{d\times d}} \mathcal{L}(\mathbf{W},\mathbf{M},\mathbf{Y}) \ , \tag{9}$$

where $\quad \mathcal{L}(\mathbf{W},\mathbf{M},\mathbf{Y}) := \frac{1}{T}\operatorname{Tr}\left(-2\mathbf{X}^\top\mathbf{W}^\top\mathbf{Y} + \mathbf{Y}^\top\mathbf{M}\mathbf{Y}\right) + \operatorname{Tr}\left(\mathbf{W}\mathbf{\Gamma}_x^{-1}\mathbf{W}^\top\mathbf{\Lambda}^{-2} - \mathbf{M}\right) \ .$

To verify the equivalence between the minimization problem (8) and the min-max problem (9) take partial derivatives of $\mathcal{L}(\mathbf{W},\mathbf{M},\mathbf{Y})$ with respect to $\mathbf{W}$ (resp. $\mathbf{M}$) and note that the minimum (resp. maximum) is achieved when $\mathbf{W} = \frac{1}{T}\mathbf{\Lambda}^2\mathbf{Y}\mathbf{X}^\top\mathbf{\Gamma}_x$ (resp. $\frac{1}{T}\mathbf{Y}\mathbf{Y}^\top = \mathbf{I}_d$). Substituting optimal $\mathbf{W}$ and $\mathbf{M}$ leads back to (8).

Finally, interchanging the order of minimization with respect to $\mathbf{Y}$, with the optimization with respect to $\mathbf{W}$ and $\mathbf{M}$, yields

$$\min_{\mathbf{W}\in\mathbb{R}^{d\times d}} \max_{\mathbf{M}\in\mathbb{R}^{d\times d}} \min_{\mathbf{Y}\in\mathbb{R}^{d\times T}} \mathcal{L}(\mathbf{W},\mathbf{M},\mathbf{Y}). \tag{10}$$

The interchange is justified by saddle point property of $\mathcal{L}(\mathbf{W},\mathbf{M},\mathbf{Y})$ with respect to $\mathbf{Y}$ and $\mathbf{M}$ [44].

## 4.2 Gradient optimization in the offline setting

In this subsection, we optimize the objective (6) in the offline setting, where the entire data matrix $\mathbf{X}$ is accessible. In this case, we solve the min-max problem (9) by alternating optimization steps. For fixed $\mathbf{W}$ and $\mathbf{M}$, we minimize the objective function $\mathcal{L}(\mathbf{W}, \mathbf{M}, \mathbf{Y})$ over $\mathbf{Y}$, which yields the relation

$$\mathbf{Y} := \underset{\mathbf{Y} \in \mathbb{R}^{d \times T}}{\arg \min} \, \mathcal{L}(\mathbf{W}, \mathbf{M}, \mathbf{Y}) \, = \, \mathbf{M}^{-1} \mathbf{W} \mathbf{X} \, . \tag{11}$$

Before applying gradient optimization steps of the objective function $\mathcal{L}(\mathbf{W}, \mathbf{M}, \mathbf{Y})$ with respect to $\mathbf{W}$ and $\mathbf{M}$, we first simplify $\boldsymbol{\Gamma}_x^{-1}$ appearing in (10), as a part of the term, $\mathrm{Tr}\left(\mathbf{W}\boldsymbol{\Gamma}_x^{-1}\mathbf{W}^\top \boldsymbol{\Lambda}^{-2}\right)$,

$$\boldsymbol{\Gamma}_x^{-1} = \mathbf{C}_x^{1/2} \left[ \frac{1}{T} \sum_{t=1}^{T} \|\mathbf{C}_x^{-1/2}\mathbf{x}_t\|^2 \mathbf{C}_x^{-1/2}\mathbf{x}_t\mathbf{x}_t^\top \mathbf{C}_x^{-1/2} \right] \mathbf{C}_x^{1/2} \; = \; \frac{1}{T} \sum_t \alpha_t \mathbf{x}_t \mathbf{x}_t^\top \, , \tag{12}$$

where $\alpha_t = \|\mathbf{C}_x^{-1/2}\mathbf{x}_t\|^2$ is the squared norm of the whitened data. As the transformation from whitened data to the recovered sources is an orthogonal rotation, the squared norm of the sources and of the outputs is preserved:

$$\alpha_t = \|\mathbf{C}_x^{-1/2}\mathbf{x}_t\|^2 = \|\mathbf{s}_t\|^2 = \|\mathbf{y}_t\|^2 \, . \tag{13}$$

We then use (13) to rewrite (12) as

$$\boldsymbol{\Gamma}_x^{-1} = \tfrac{1}{T}\mathbf{X}\,\mathrm{ddiag}(\mathbf{Y}^\top \mathbf{Y})\mathbf{X}^\top = \tfrac{1}{T} \sum_{t=1}^{T} \|\mathbf{y}_t\|^2 \mathbf{x}_t \mathbf{x}_t^\top, \tag{14}$$

where $\mathrm{ddiag}(\cdot)$ represents a diagonal matrix which keeps only the diagonal elements of the argument matrix.

We now obtain the update rules for $\mathbf{W}$ and $\mathbf{M}$ by gradient-descent ascent on (10) and by replacing $\boldsymbol{\Gamma}_x^{-1}$ according to (14)

$$\mathbf{W} \leftarrow \mathbf{W} + 2\eta\left(\tfrac{1}{T}\mathbf{Y}\mathbf{X}^\top - \boldsymbol{\Lambda}^{-2}\mathbf{W}\boldsymbol{\Gamma}_x^{-1}\right) = \mathbf{W} + \frac{2\eta}{T}\left(\mathbf{Y}\mathbf{X}^\top - \boldsymbol{\Lambda}^{-2}\mathbf{W}\mathbf{X}\,\mathrm{ddiag}(\mathbf{Y}^\top \mathbf{Y})\mathbf{X}^\top\right) \, , \tag{15}$$

$$\mathbf{M} \leftarrow \mathbf{M} + \frac{\eta}{\tau}\left(\frac{1}{T}\mathbf{Y}\mathbf{Y}^\top - \mathbf{I}_d\right) \, . \tag{16}$$

Here $\tau > 0$ is the ratio between the learning rates for $\mathbf{W}$ and $\mathbf{M}$, and $\eta \in (0, \tau)$ is the learning rate for $\mathbf{W}$, ensuring that $\mathbf{M}$ remains positive definite given a positive definite initialization.

## 4.3 Online algorithm

We now solve the min-max objective (9) in the online setting. At each time step, $t$, we minimize over the output, $\mathbf{y}_t$, by repeating the following gradient descent steps until convergence:

$$\mathbf{y}_t \leftarrow \mathbf{y}_t + \gamma(\mathbf{c}_t - \mathbf{M}\mathbf{y}_t) \, , \tag{17}$$

where $\gamma$ is a small step size, and we have defined the projection $\mathbf{c}_t := \mathbf{W}\mathbf{x}_t$, with biological interpretation described in Sec 5. As in (11), the dynamics converge to $\mathbf{y}_t = \mathbf{M}^{-1}\mathbf{c}_t$. We now take stochastic gradient descent-ascent steps in $\mathbf{W}$ and $\mathbf{M}$. We thus replace the averages in Eqs. (15)-(16) with their online approximations

$$\frac{1}{T}\mathbf{Y}\mathbf{X}^\top \rightarrow \mathbf{y}_t\mathbf{x}_t^\top \; ; \; \frac{1}{T}\mathbf{Y}\mathbf{Y}^\top \rightarrow \mathbf{y}_t\mathbf{y}_t^\top \; ; \; \frac{1}{T}\boldsymbol{\Lambda}^{-2}\mathbf{W}\mathbf{X}\,\mathrm{ddiag}(\mathbf{Y}^\top \mathbf{Y})\mathbf{X}^\top \rightarrow \|\mathbf{y}_t\|^2\boldsymbol{\Lambda}^{-2}\mathbf{c}_t\mathbf{x}_t^\top \, .$$

This yields our online ICA algorithm (Algorithm 1) and the NN, see Section 5.

## 5 Biological interpretation and neural implementation

We now show that our online ICA algorithm (Algorithm 1) maps onto an NN with local, activity-dependent synaptic update rules, which emulate aspects of synaptic plasticity observed experimentally.

---

**Algorithm 1** A similarity-preserving algorithm for Independent Component Analysis.

---

**input** data $\{\mathbf{x}_1, \ldots, \mathbf{x}_T\}$; dimension $d$
**output** $\{\mathbf{y}_1, \ldots, \mathbf{y}_T\}$; dimension $d$                          ▷ estimated sources
**initialize** the matrix $\mathbf{W}$, and positive definite matrix $\mathbf{M}$.
**for** $t = 1, 2, \ldots, T$ **do**
   $\mathbf{c}_t \leftarrow \mathbf{W}\mathbf{x}_t$ ;                          ▷ projection of inputs
   **run the following until convergence:**
     $\frac{d\mathbf{y}_t(\gamma)}{d\gamma} = \mathbf{c}_t - \mathbf{M}\mathbf{y}_t(\gamma)$ ;                          ▷ neural dynamics
   $\mathbf{W} \leftarrow \mathbf{W} + 2\eta(\mathbf{y}_t - \|\mathbf{y}_t\|^2\mathbf{\Lambda}^{-2}\mathbf{c}_t)\mathbf{x}_t^\top$ ; $\mathbf{M} \leftarrow \mathbf{M} + \frac{\eta}{\tau}(\mathbf{y}_t\mathbf{y}_t^\top - \mathbf{I}_d)$ ; ▷ synaptic updates
**end for**

---

## 5.1 Neural architecture and dynamics

Our algorithm can be implemented by a biologically plausible NN presented in **Fig. 1B**. The network consists of an input layer of $d$ neurons, representing the input data to be separated into independent components, and an output layer of $d$ neurons, with separate dendritic and somatic compartments, estimating the unknown sources. The network includes a set of feedforward synapses between the inputs and the dendrites of the output neurons as well as a set of lateral synapses between the output somas **Fig. 1B**.

Although two-compartment neurons have not been common in machine learning, in neuroscience, they are often used to model pyramidal cells - the most numerous neuron type in the neocortex. Such neuron consists of an apical dendritic compartment, as well as a somatic compartment, which have distinct membrane potentials [56, 57, 58, 59]. Recently, such two-compartment neurons appeared in bio-inspired machine learning algorithms [60, 61, 62, 51, 63, 64].

At each time step $t$, the network computes in two phases. First, the mixture $\mathbf{x}_t$, represented in the input neurons, is multiplied by the weight matrix $\mathbf{W}$ encoded by the feedforward synapses connecting the input neurons to the output neurons. This yields the projection $\mathbf{c}_t = \mathbf{W}\mathbf{x}_t$ computed in the dendritic compartments of the output neurons and then propagated to their somatic compartments.

Second, the $d$-dimensional output signal $\mathbf{y}_t$ is computed as somatic activity in the output neurons and corresponds to the estimated sources. This is accomplished by the fast recurrent neural dynamics in lateral connection, Eq. (17), converging to the equilibrium value assignment of $\mathbf{y}_t$ in Algorithm 1. The lateral synapses in **Fig. 1B** implement only the off-diagonal elements of $\mathbf{M}$. Whereas diagonal elements of $\mathbf{M}$ would correspond to autapses (self-coupling of neurons), such Hebbian/ani-Hebbian networks can be designed without them [44, 45].

## 5.2 Synaptic plasticity rules

To highlight the locality of our learning rules, we rewrite the element-wise synaptic updates for $\mathbf{W}$ and $\mathbf{M}$ in Algorithm 1 using sub-/super-scripts:

$$W_{ij} \leftarrow W_{ij} + 2\eta \left( y_t^i x_t^j - \|\mathbf{y}_t\|^2 \frac{c_t^i}{\lambda_i^2} x_t^j \right) \ ; \ M_{ij} \leftarrow M_{ij} + \frac{\eta}{\tau} \left( y_t^i y_t^j - \delta_{ij} \right), \ 1 \leq i, j \leq d . \quad (18)$$

In Eqs. (18), $x_t^j$ is the activity of the $j^{th}$ input neuron, $y_t^i$ is the activity of the $i^{th}$ output neuron, and $c_t^j$ is the dendritic current of the $j^{th}$ output neuron, all at time $t$. Furthermore, the influence of the dendritic current $c_j^t$ on a synapse's strength is modulated by the term $\|\mathbf{y}_t\|^2$, representing the overall activity of the output neurons.

How could total output neuronal activity be signaled to each feedforward synapse in the network? There are several diffusible molecules in the brain which may affect synaptic plasticity and whose concentration may depend on the overall neural activity. These include extracellular calcium [33], GABA[29, 65], dopamine [30, 66, 67], noradrenaline [31, 68], D-Serin [32, 69, 70] or nitric oxide (NO), although its range of action is contested [35, 36, 37]. Finally, local field potential can also affect synaptic plasticity [34]. For the learning rule to function in the online setting signaling must be fast, a requirement favoring local field potential and extracellular calcium out of the above candidates.

The learning rule (18) for feedforward synaptic weights, $\mathbf{W}$, simplifies significantly near the optimum of the objective, $\mathbf{M} \approx \mathbf{I}_d$, and for the converged output activity, $\mathbf{c}_t = \mathbf{W}\mathbf{x}_t = \mathbf{M}\mathbf{y}_t$:

$$\Delta\mathbf{W} = \eta_t \left(\mathbf{I} - \|\mathbf{y}_t\|^2 \mathbf{\Lambda}^{-2}\mathbf{M}\right) \mathbf{y}_t\mathbf{x}_t^\top \approx \eta_t \left(\mathbf{I} - \|\mathbf{y}_t\|^2 \mathbf{\Lambda}^{-2}\right) \mathbf{y}_t\mathbf{x}_t^\top \ . \tag{19}$$

Such an update is a nonlinearly modulated Hebbian learning rule where the sign of plasticity changes with the total output activity. For low total output activity, $\|\mathbf{y}_t\|^2 < \lambda_i^2$, the update is Hebbian, i.e., long-term potentiation (LTP) for correlated inputs and outputs. For high output activity, $\|\mathbf{y}_t\|^2 > \lambda_i^2$, the update is anti-Hebbian, i.e., long-term depression (LTD) for correlated inputs and outputs. We compare and contrast this global activity-dependent modulation of plasticity with the Bienenstock, Cooper, and Munro (BCM) rule in subsection 5.3.

Whereas three-factor learning has been invoked in multiple computational, especially reward-based, models [71, 72, 73, 74], our model is the first to propose such learning in the fully normative approach for ICA.

## 5.3   Comparison with existing rules

To understand the distinctive features of our model versus existing approaches, we compare and contrast it with three existing models: 1. Oja's learning rule, 2. BCM learning rule [75], and 3. error-gated Hebbian rule (EGHR) [69].

**1. Nonlinear Oja's learning rule.** [76, 19] generalized the original Oja's rule, Eq. (3), with a component-wise nonlinear function $g(\cdot)$ as $\Delta\mathbf{W} = g(\mathbf{y}_t)\mathbf{x}_t^\top - g(\mathbf{y}_t)g(\mathbf{y}_t)^\top\mathbf{W}$. However, this model and follow-up work inherited the main drawbacks of the standard Oja's rule, i.e., they require pre-whitening of the data and rely on non-local learning rules. Indeed, the last term of the learning rules of nonlinear Oja implies that updating the weight of a synapse requires precise knowledge of output activities of all other neurons which are not available to the synapse (cf. [77] for details on standard PCA rules and networks of nonlinear neurons [78]).

**2. BCM learning rule.** Switching of the sign of plasticity depending on the total output activity Eq. (19) is reminiscent of the BCM rule. It was initially postulated and later connected to an objective function [79, 80] characterizing the deviation from Gaussian distribution but mainly focusing on skewness rather than kurtosis as in our model. For correlated input and output, the BCM rule induces LTD for "sub-threshold" responses and LTP for "super-threshold" responses, with the threshold being a function of average output activity. Unfortunately, multiple output BCM neurons respond to the same dominant feature, producing an incomplete, highly redundant code [79, 80]. In contrast, our network has lateral inhibitory connections whose weights are updated via anti-Hebbian rule Eq. (18) leading to the recovery of multiple sources. Although experimental evidence has validated BCM-like plasticity in parts of the visual cortex and the hippocampus, other brain areas have yet to show similar behavior. Interestingly, an "inverse" BCM rule, similar to ours has been proposed in the cerebellum [81, 82, 83].

**3. Modulated Hebbian rules.** Recent neural implementations of ICA [69, 74, 70] also introduced modulated Hebbian rules: $\Delta\mathbf{W} = (E_0 - E(\mathbf{y}_t))g(\mathbf{y}_t)\mathbf{x}_t^\top$, with $E_0$ a constant, $E(\cdot)$ a nonlinear function of the total activity, and $g(\cdot)$ a component-wise nonlinear function. This learning rule shares many similarities with ours. The term $E_0$ is a constant characterizing the source distributions, which could identify with our $\lambda_i$ terms, and the function $E(\cdot)$ resembles our $\|\mathbf{y}_t\|^2$ but is model dependent in their approach. This is where the similarities end as their objective function is inspired by the information-theoretic framework [12, 7] and ours - by the insight from the FOBI procedure [27, 84] and spectral methods from the SM method [44].

Their model can be considered partly normative since the neural architecture is predetermined and uses a hand-designed error-computing neuron to determine the global modulating factor rather than having been derived from an optimization problem. Interestingly, their model does not use lateral connections for output decorrelation resulting in a model without direct interaction between outputs. The presence of pairwise inhibitory interaction is crucial for our algorithm, leading to a globally optimal solution when the sources have distinct kurtosis. Numerical and theoretical analysis of the performance of the EGHR algorithm relies on the source distributions being the same and resulting in several equivalent optima.

# 6 Numerical simulations

In this section, we verify our theoretical results in numerical experiments. We use our model to perform ICA on both synthetic and real-world datasets. We designed three sets of experiments to illustrate the performance of our algorithm. In the first set, **Fig. 2A**, we used as sources artificially generated signals, in the second - natural speech signals, **Fig. 2B**, and in the third - natural scene images, **Fig. 2C**. According to the generative model, Eq. (1), we then used random full rank square mixing matrices, $\mathbf{A}$, to generate the observed mixed signals, $\mathbf{x}_t$. From $\mathbf{x}_t$, we aimed to recover the original sources. We show that our algorithm recovers sources regardless of sub- or super-Gaussianity of the kurtosis, which is essential for natural datasets. For details on the parameters used, see Appendix C.

**Synthetic data.** We first evaluate our algorithm on a synthetic dataset generated by independent and identically distributed samples. The data are generated from periodic signals, i.e., square-periodic, sine-wave, saw-tooth, and Laplace random noise. The data were chosen with the purpose of including both super- and sub-Gaussian distribution known respectively as leptokurtic ("spiky", e.g., the Laplace distribution) and platykurtic ("flat-topped", the three other source signal). We show in **Fig. 2A** the mixed signals in black, on the left plots. We show on the right plot the recovered sources, in red, overlapped with the original sources, in blue, and the residual in green. We also show the histogram of each signal on the right side of each plot. Results are shown for 300 samples. We observe that the recovered and true sources nearly perfectly overlap, explaining the low value of the residual, which shows the almost perfect reconstruct performed by our algorithm.

In Appendix **D**, we provide a numerical comparison of the performance of our algorithm with competing models, namely, Herault-Jutten algorithm [85], EASI algorithm [86, 87], Bell and Sejnowski's algorithm [12], the Amari algorithm [13], and finally nonlinear Oja algorithm [88, 76]. These models were designed with NNs in mind and are seminal works on neural ICA algorithms. However, like nonlinear Oja algorithm, which is mentioned in Section 5.3.1, these models suffer from biological implausibility. In brief, our model either outperforms or is competitive with the models mentioned above. These results also confirm that our algorithm can deal with combinations of sub- and super-Gaussian sources, with or without pre-whitening of the data.

**Real-world data: Speech signals.** For the audio separation task, we used speech recordings from the freely available TSP data set [89][1] , recorded at 16kHz. The first source we use was obtained from a male speaker (MA02 04.wav), the second source from a female speaker (FA01 03.wav), and the third source synthetically generated from a uniform noise, as was previously used in the literature [90]. We show our results in **Fig. 2B**. We show the mixtures, the true sources, the recovered sources, and the residual. It is clear from the figure that our algorithm's outputs recover the true sources similarly to the synthetic dataset.

**Real-world data: Natural scene images.** We finally applied our algorithm to the task of recovering images from their mixtures, on data already used for BSS tasks [7, 91, 23] [2], as shown in **Fig. 2C**. Here, we show separately the original sources, top images, the mixtures, middle images, and the recovered sources, bottom images of **Fig. 2C**. We considered three grayscale images of size $256 \times 512$ pixels (shifted and scaled to have zero-mean and unit variance), such that each image is treated as one source, with the pixel intensities representing the samples. We again observe in **Fig. 2C** that the recovered sources are nearly identical to the original sources. We can also see that the histograms of the recovered sources nearly match the histograms of the original sources, up to their sign.

# 7 Discussion

We proposed a new single-layer ICA NN with biologically plausible local learning rules. The normative nature of our approach makes the biologically realistic features of our NNs readily interpretable. In particular, our NN uses neurons with two separate compartments and is trained with extended Hebbian learning rules. The changes in synaptic strength are modulated by the total output neuronal activity, equivalent to performing gradient optimization of our objective function. We demonstrated that the proposed rule reliably converges to the correct solution over a wide range of mixing matrices, synthetic, and natural datasets. The broad applicability and easy implementation

---

[1]Freely available at http://www.mmsp.ece.mcgill.ca/Documents/Data/ (Accessed May 24th 2021).

[2]Freely available at https://research.ics.aalto.fi/ica/data/images/ (Accessed May 24th 2021).

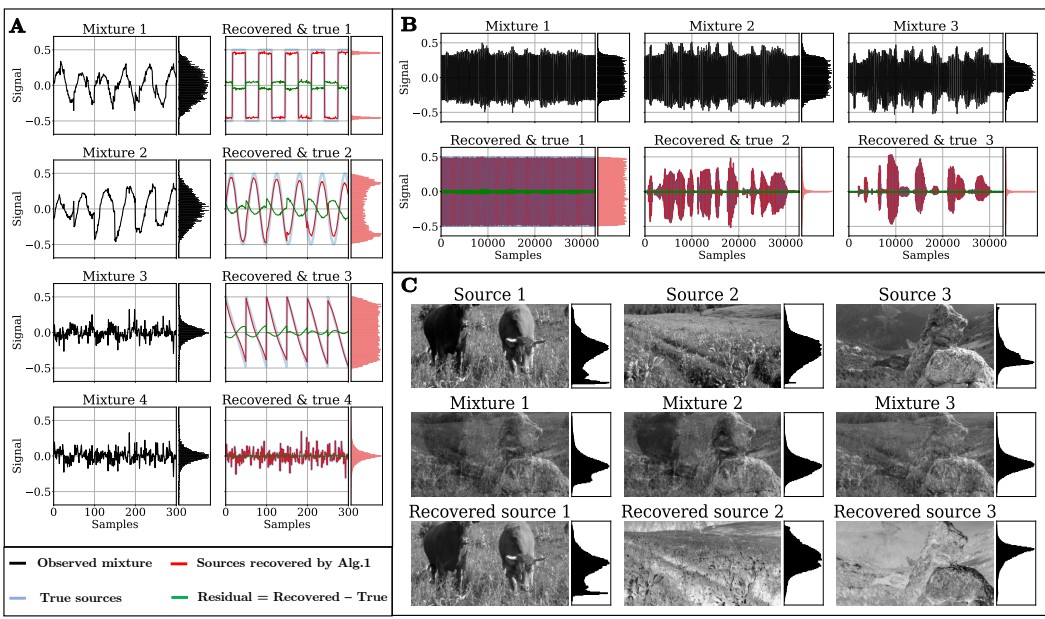

Figure 2: **Our ICA algorithm recovers independent sources from synthetic and real-world mixtures. A.** Synthetic data, **B.** Natural speech data, **C.** Natural image data. In **A-B.** mixed signals are shown in black, the recovered signals - in red, the true sources - in blue, and their residual difference - in blue. We also show the associated distributions. **C.** The sources, mixture, and recovered sources, in top, middle and bottom rows respectively.

of our NN and learning rules could further advance neuromorphic computation [92, 93, 94] and may reveal the principle underlying BSS computation in the brain.

Recent work on canonical correlation analysis [95, 96, 97], slow feature analysis [98], and ICA-like algorithms have led to biologically plausible NNs [99, 100, 101, 102], some of which rely on two-compartment neurons [103, 51]. These NNs could, in principle, be used for popular BSS tasks known as second-order blind identification [104, 105, 106, 107] or in the context of kernel ICA [108, 109]. This suggests the existence of a single model of two-compartment neurons and non-trivial local learning rules for BS. In future work, we aim at proposing such a model, including high-order statistics, temporal correlation, and diversity of views.

One limitation of our approach is the inability of the model to separate sources with the same kurtosis. Yet, as long as sources possess some distinct even-order moments, our scaling rule can be altered to separate the sources [27]. Another limitation is the well-known sensitivity of kurtosis to outliers. This limitation could be overcome if scaling varies as a sublinear function of the total activity [40]. These changes do not affect the neural architecture nor the locality of the learning rules.

Clarifying the limitations of our model leads us to ask various follow-up questions left for future work. How can we further generalize the solution beyond the choice of nonlinearity and beyond the task of linear ICA? We could envision considering more than two covariance matrices as in the JADE algorithm [110, 111, 112], which effectively performs joint-diagonalization of arbitrarily many matrices. A neural solution was proposed in [113] but again relies on non-local Oja-based rules. Ongoing work on nonlinear ICA [114, 115] is of great interest to us since it might be a perfect candidate for multi-layered architectures.

Recently, several works proposed biologically plausible supervised learning algorithms [61, 62, 63, 53, 116]. Combining these with ICA and unsupervised learning algorithms in general would provide a more comprehensive description of cognitive processes.

## Acknowledgments and Disclosure of Funding

Y.B. is grateful to Romain Cosentino, and Claudia Skok Gibbs for insightful discussions related to this work and feedback on this manuscript. We also thank the members of the Neural Circuits and Algorithms Group at the Flatiron Institute for providing feedback on an early version of this work.

The authors did not receive any third party funding for the completion of this project.

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
