This is the supplementary material for the NeurIPS titled "A Normative and Biologically Plausible Algorithm for Independent Component Analysis", from Yanis Bahroun, Dmitri B. Chklovskii, and Anirvan M. Sengupta.

# A  Proof of our main theorem

We start by restating the setup in which our algorithm operates. The type of ICA considered in our work assumes the following generative model. There are $d$ sources recorded $T$ times forming the columns of $\mathbf{S} := [\mathbf{s}_1, \ldots, \mathbf{s}_T] \in \mathbb{R}^{d \times T}$ whose components $s_t^1, \ldots, s_t^d$ are assumed non-Gaussian and independent. Without loss of generality, we assume that each source has zero-mean, unit variance, and finite and distinct kurtosis, a common assumption among kurtosis-based ICA methods [12]. The kurtosis of a random variable $v$ is defined as $\mathrm{kurt}[v] = \mathbb{E}\left[(v - \mathbb{E}(v))^4\right] / \left(\mathbb{E}\left[(v - \mathbb{E}(v))^2\right]\right)^2$. Finally, sources are assumed to be mixed through a linear system, i.e., there exists a full rank mixing matrix, $\mathbf{A} \in \mathbb{R}^{d \times d}$, producing the $d$-dimensional mixture, $\mathbf{x}_t$, expressed as

$$\mathbf{x}_t = \mathbf{A}\mathbf{s}_t \qquad \forall t \in \{1, \ldots, T\} \ . \tag{S.1}$$

The goal of ICA algorithms is then to determine a signal, $\mathbf{y}_t$, obtained from a fixed linear transformation of the observed signal, $\mathbf{x}_t$, i.e., $\exists \mathbf{W}_{ICA} \in \mathbb{R}^{d \times d}$, such that

$$\mathbf{y}_t = \mathbf{\Xi}\mathbf{\Pi}\mathbf{s}_t \quad \text{and} \quad \mathbf{y}_t := \mathbf{W}_{ICA}\mathbf{x}_t, \quad \forall t \in \{1, \ldots, T\} \ , \tag{S.2}$$

where $\mathbf{\Xi}$ is a diagonal matrix with $\pm 1$'s on the diagonal, and $\mathbf{\Pi}$ a permutation matrix. As a result, $\mathbf{y}_t$ represents the ideally recovered unknown sources.

We now recall our objective function for ICA

$$\min_{\substack{\mathbf{Y} \in \mathbb{R}^{d \times T} \\ \frac{1}{T}\mathbf{Y}\mathbf{Y}^\top = \mathbf{I}_d}} \left\| \mathbf{X}^\top \mathbf{C}_x^{-1/2} \left[ \frac{1}{T} \sum_{t=1}^{T} \|\mathbf{C}_x^{-1/2}\mathbf{x}_t\|^2 \mathbf{C}_x^{-1/2}\mathbf{x}_t\mathbf{x}_t^\top \mathbf{C}_x^{-1/2} \right]^{-1} \mathbf{C}_x^{-1/2}\mathbf{X} - \mathbf{Y}^\top \mathbf{\Lambda}^2 \mathbf{Y} \right\|_F^2 \ . \tag{S.3}$$

and finally our main theorem which is proved in the following.

**Theorem 1.** *Given that the sources are independent, centered, have unit variance, and distinct kurtosis (c.f. Sec 2 of the main text), then the global optimal solution for our objective* (S.3), *denoted by* $\mathbf{Y}^*$ *satisfies*

$$\mathbf{Y}^* = \mathbf{\Xi}\mathbf{\Pi}\mathbf{S} \tag{S.4}$$

*where* $\mathbf{\Xi}$ *and* $\mathbf{\Pi}$, *defined in* (S.2), *represents the sign and permutation ambiguity of the solution, and is thus a solution to the ICA problem.*

We propose an outline of the proof.

- Proof of the FOBI procedure
- Proof that $\mathbf{C}_z$ and $\mathbf{C}_z^{-1}$ lead to the same solution of the eigenvalue problem solving FOBI.
- Note that $\mathbf{C}_z^{-1}$ and $\mathbf{C}_z^{-1/2}\mathbf{C}_h\mathbf{C}_z^{-1/2}$ are equivalent by whitening properties.
- Identify the Gramian associated with $\mathbf{C}_z^{-1/2}\mathbf{C}_h\mathbf{C}_z^{-1/2}$, which is $\mathbf{H}^\top \mathbf{C}_z^{-1}\mathbf{H}$.
- Use the Similarity matching formulation of principal component analysis on the aforementioned Gramian to finalize the proof.

## A.1  Statements and Proofs of FIBO

We redefine important notations, i.e., the whitened data denoted by $\mathbf{h}_t$ is defined as

$$\mathbf{h}_t = \mathbf{C}_x^{-1/2}\mathbf{x}_t \ .$$

Now the weighted whitened data, scaled by its norm, denoted by $\mathbf{z}_t$, is defined as

$$\mathbf{z}_t = \|\mathbf{h}_t\| \cdot \mathbf{h}_t \ .$$

We have defined the sample covariance matrix of $\mathbf{x}_t$ and $\mathbf{z}_t$ as $\mathbf{C}_x$ and $\mathbf{C}_z$ respectively.

Before, demonstrating our theorem, we need to demonstrate the main results of the FOBI procedure, i.e., we must show that when the whitened data, $\mathbf{h}_t$ are projected onto the eigenvectors of the weighted data, $\mathbf{z}_t$ they recover the sources. This results is stated and proved using the following theorem from [4, 22].

**Theorem 2.** *Let $\mathbf{s}$, $\mathbf{h}$, and $\mathbf{z}$ be random vectors such that $\mathbf{h} = \mathbf{B}\mathbf{s}$, where $\mathbf{B}$ is an orthogonal matrix, and $\mathbf{z} = \|\mathbf{h}\| \cdot \mathbf{h}$. Suppose additionally $\mathbf{s}$ has zero-mean independent components and these components have distinct finite kurtosis. Then the orthogonal matrix $\mathbf{U}$ which gives the principal components of $\mathbf{z}$ performs ICA on $\mathbf{h}$.*

*Proof.* Let $\mathbf{s} = [s_1, \ldots, s_d]^\top$, $\mathbf{U} = [\mathbf{u}_1, \ldots, \mathbf{u}_d]$, and $\mathbf{C} = [\mathbf{c}_1, \ldots, \mathbf{c}_d] = [c_{ij}]_{d \times d} = \mathbf{U}\mathbf{B}$. Since $\mathbf{B}$ is orthogonal, we have $\|\mathbf{h}\| = \|\mathbf{B}\mathbf{s}\| = \|\mathbf{s}\|$. The second-order moment of the projection of $\mathbf{z}$ on $\mathbf{u}_i$ is

$$\mathbb{E}(\mathbf{u}_i^\top \mathbf{z})^2 = \mathbb{E}(\mathbf{u}_i^\top \|\mathbf{h}\| \cdot \mathbf{h})^2 = \mathbb{E}(\|\mathbf{s}\|^2 \cdot (\mathbf{u}_i^\top \mathbf{B}\mathbf{s})^2) = \mathbb{E}\left(\sum_{k=1}^d (\mathbf{c}_i^\top \mathbf{s})^2 \cdot s_k^2\right)$$

$$= \sum_{k=1}^d c_{ik}^2 \mathbb{E}(s_k^4) + \sum_{k=1}^d \sum_{p=1, p \neq k}^d c_{ip}^2 \mathbb{E}(s_p^2 s_k^2) + \sum_{k=1}^d \sum_{p=1, p \neq k}^d \sum_{q=1, q \neq p}^d c_{ip} c_{iq} \mathbb{E}(s_p s_q s_k^2) \tag{S.5}$$

When $q \neq p$, at least one of $q$ and $p$ is different from $k$. Suppose $q \neq k$. We then have $\mathbb{E}(s_p s_q s_k^2) = \mathbb{E}(s_q)\mathbb{E}(s_p s_k^2) = 0$ since $s_i$ are independent and zero-mean. We also have $\sum_{k=1}^d c_{ik}^2 = 1$ since $\mathbf{C}$ is orthogonal. Equation (S.5) then becomes

$$\mathbb{E}(\mathbf{u}_i^\top \mathbf{z})^2 = \sum_{k=1}^d c_{ik}^2 \mathbb{E}(s_k^4) + \sum_{k=1}^d \sum_{p=1, p \neq k}^d c_{ip}^2 \mathbb{E}(s_p^2 s_k^2)$$

$$= \sum_{k=1}^d c_{ik}^2 \mathbb{E}(s_k^4) + \sum_{k=1}^d \sum_{p=1, p \neq k}^d c_{ip}^2$$

$$= \sum_{k=1}^d c_{ik}^2 \, \mathrm{kurt}(s_k) + d + 2 \tag{S.6}$$

Therefore $\mathbb{E}(\mathbf{u}_i^\top \mathbf{z})^2$ is the weighted average of $kurt(s_i)$ plus a constant. As $s_i$ are assumed to have different kurtosis, without loss of generality, we assume $kurt(s_1) > kurt(s2) > \cdots > kurt(s_n)$. From Equation (S.6) we can see that maximization of $\mathbb{E}(\mathbf{u}_1^\top \mathbf{z})^2$ gives $\mathbf{c}_1 = [\pm 1, 0, \ldots, 0]^\top$, which means that $y_1 = \mathbf{u}_1^\top \mathbf{h} = \mathbf{u}_1^\top \mathbf{B}\mathbf{s} = \mathbf{c}_1^\top \mathbf{s} = \pm s_1$. After finding $\mathbf{u}_1$, under the constraint that $\mathbf{u}_2$ is orthogonal to $\mathbf{u}_1$, the maximum of $\mathbb{E}(\mathbf{u}_2^\top \mathbf{z})^2$ is obtained at $\mathbf{c}_2 = [0, \pm 1, 0, \ldots, 0]^\top$. Consequently $y_2 = \mathbf{u}_2^\top \mathbf{h} = \mathbf{c}_2^\top \mathbf{s} = \pm s_2$. Repeating this procedure, finally all independent components can be estimated as $y_i = \mathbf{u}_i^\top \mathbf{h} = \pm s_i$ where $\mathbf{u}_i$ maximizes $\mathbb{E}(\mathbf{u}_i^\top \mathbf{z})^2$. In other words, the orthogonal matrix $\mathbf{U}$ performing PCA on $\mathbf{z}$ (without centering of $\mathbf{z}$) actually performs ICA on $\mathbf{h}$. $\qquad \square$

We have thus proved that the information about the eigenvectors of $\mathbf{z}_t$ are enough for finding the sources after whitening.

## A.2 Eigenvalue Formulation

Now that we have clarified that the FOBI procedure consists of projecting the input data using Theorem 2, we can rephrase it as a generalized eigenvalue problem where $\mathbf{W}$ is the solution of the following problem

$$\mathbf{W}\mathbf{W}^\top = \mathbf{I} \text{ and } \mathbf{W}\mathbf{C}_z\mathbf{W} = \mathbf{D} , \tag{S.7}$$

with $\mathbf{D}$ a diagonal matrix with distinct nonnegative diagonal elements. This problem is equivalent to

$$\tilde{\mathbf{W}}\tilde{\mathbf{W}}^\top = \mathbf{I} , \quad \tilde{\mathbf{W}}\mathbf{C}_z^{-1}\tilde{\mathbf{W}}^\top = \mathbf{D}^{-1} , \tag{S.8}$$

With $\mathbf{H}$ and $\mathbf{h}_t$, $\mathbf{Z}$ and $\mathbf{z}_t$ define as in the main text. This results from the fact that $\mathbf{C}_z$ is assumed full rank and have distinct eigenvalue, the problem $\mathbf{W}\mathbf{C}_z\mathbf{W} = \mathbf{D}$ has the same solution as $\mathbf{W}\mathbf{C}_z^{-1}\mathbf{W} = \mathbf{D}^{-1}$ under orthogonality constraint.

Now we note that by definition $\mathbf{C}_h = \frac{1}{T}\mathbf{H}\mathbf{H}^\top = \mathbf{I}_d$, and that $\mathbf{C}_z^{-1}$ is positive semi-definite thus admits a square-root. We can then rephrase the problem (S.8) as

$$\underset{\hat{\mathbf{W}}^\top\hat{\mathbf{W}}=\mathbf{I}_d}{\arg\max}\ \mathrm{Tr}(\hat{\mathbf{W}}^\top\mathbf{C}_z^{-1/2}\mathbf{C}_h\mathbf{C}_z^{-1/2}\hat{\mathbf{W}}) = \underset{\hat{\mathbf{W}}^\top\hat{\mathbf{W}}=\mathbf{I}_d}{\arg\max}\ \mathrm{Tr}(\hat{\mathbf{W}}^\top\mathbf{C}_z^{-1/2}\frac{1}{T}\mathbf{H}\mathbf{H}^\top\mathbf{C}_z^{-1/2}\hat{\mathbf{W}}) \quad (\text{S.9})$$

### A.3 From the covariance matrix to the Gram matrix

In the final step we need to consider the following matrix $\mathbf{K} = \mathbf{H}\mathbf{C}_z^{-1/2}$, which is a simple but essential rewriting of the eigenvalue problem (S.9) as

$$\underset{\hat{\mathbf{W}}\hat{\mathbf{W}}^\top=\mathbf{I}_d}{\arg\max}\ \mathrm{Tr}(\hat{\mathbf{W}}\mathbf{K}\mathbf{K}^\top\hat{\mathbf{W}}^\top)$$

Now we really on the result from Mardia et al.[11] for multidimensional scaling. Their results connects the eigenvalue and eigenvectors of the Gram matrix, i.e., matrix of similarity with that of the covariance matrix. This result was recently popularized by the similarity matching framework [17, 20] that states that the principal subspace projection can equivalently be obtained by

$$\underset{\tilde{\mathbf{Y}}}{\arg\min}\ \|\mathbf{K}^\top\mathbf{K} - \tilde{\mathbf{Y}}^\top\tilde{\mathbf{Y}} = \|_F^2 = \underset{\tilde{\mathbf{Y}}}{\arg\min}\ \|\mathbf{H}^\top\mathbf{C}_z^{-1}\mathbf{H} - \tilde{\mathbf{Y}}^\top\tilde{\mathbf{Y}}\|_F^2$$

However this solution does the projection onto the principal subspace but does not recover the perfect sources as it is invariant by rotation which is a problem for ICA. Thus we include $\mathbf{\Lambda}^2$ to break the symmetry as was done in [13]. The resulting objective function then leads to the same solution as the FOBI procedure.

$$\underset{\substack{\mathbf{Y}\in\mathbb{R}^{d\times T}\\ \frac{1}{T}\mathbf{Y}\mathbf{Y}^\top=\mathbf{I}_d}}{\arg\min}\ \|\mathbf{H}^\top\mathbf{C}_z^{-1}\mathbf{H} - \mathbf{Y}^\top\mathbf{\Lambda}^2\mathbf{Y}\|_F^2$$

which is exactly our objective function (S.3)

$$\underset{\substack{\mathbf{Y}\in\mathbb{R}^{d\times T}\\ \frac{1}{T}\mathbf{Y}\mathbf{Y}^\top=\mathbf{I}_d}}{\arg\min}\ \left\| \mathbf{X}^\top\mathbf{C}_x^{-1/2}\left[\frac{1}{T}\sum_{t=1}^{T}\|\mathbf{C}_x^{-1/2}\mathbf{x}_t\|^2\mathbf{C}_x^{-1/2}\mathbf{x}_t\mathbf{x}_t^\top\mathbf{C}_x^{-1/2}\right]^{-1}\mathbf{C}_x^{-1/2}\mathbf{X} - \mathbf{Y}^\top\mathbf{\Lambda}^2\mathbf{Y} \right\|_F^2 .$$

thus concluding the proof of Theorem 1.

## B Details of the illustrative example

We present the parameters used for our illustrative example for FOBI. We use a sinusoid waveform and a sawtooth signal as the independent sources. Each data source was then shuffled to remove possible temporal correlation, leading to fake dependence. We showed in Fig.**1B** 5000 datapoints. The mixing matrix $\mathbf{A}$ was randomly chosen, and in the example is

$$\mathbf{A} = \begin{bmatrix} 0.10054428 & 0.81736508 \\ 0.75216771 & 0.44640104 \end{bmatrix}$$

It is then the observed data that are shown as $\mathbf{x} = \mathbf{A}\mathbf{s}$. The rest is described in the main text. The code used to produce the figure can be found attached to the submission, `NeurIPS_Fig_1A_procedure.ipynb`.

## C Numerics

### C.1 Detailed experimental figures

We propose zoomed in version of Fig.2B and 2C of the main showing respectively the speech separation and image separation tasks. We show in Fig. 1 the results obtained by our offline algorithm

on the speech separation tasks as reported in the main text. We show in Fig. 2 the results obtained by our offline algorithm on the image separation tasks as reported in the main text. The codes used to produce the respective figures can be found attached to the submission, `NeurIPS_Fig_2B_audio`, `NeurIPS_Fig_2C_image`.

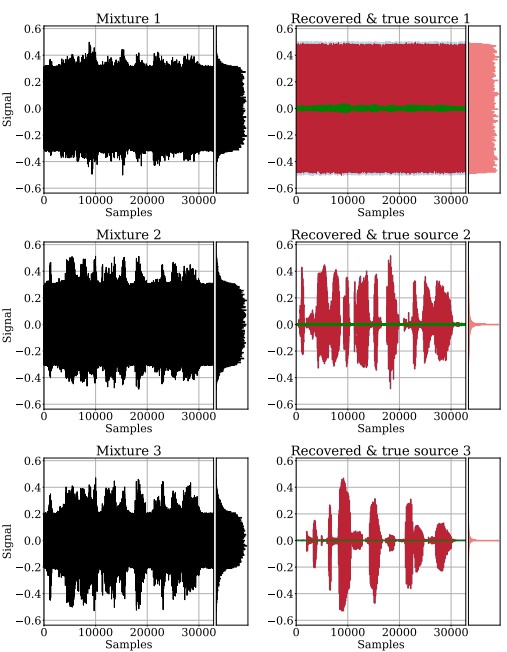

Figure 1: Illustration of the performance of our algorithm on the speech separation task. Our algorithm recovers the sources from mixed signals. It shows a zoomed in version of the results obtained for the speech separation task Fig.2.B of the main text. We show in black the mixed signals and in red (resp. blue) the recovered (resp. true) sources, and in green their difference called Residual. We also show the histogram of the associated distributions.

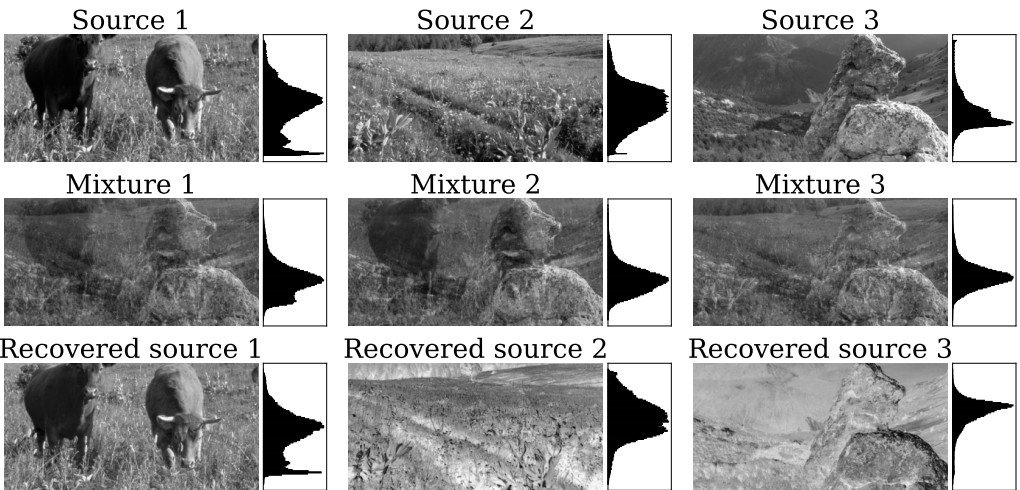

Figure 2: Illustration of the performance of our algorithm on the image separation task. Our algorithm recovers the sources from mixed images. It shows a zoomed in version of the results obtained for the image separation task Fig.2.C of the main text. We show in each row respectively the original sources, the mixed images, and the recovered alongside their histograms.

## C.2 Online algorithm results

We show in Fig. 3 the results obtained by our online algorithm as defined in the main text. We also show in Fig. 4 the results obtained by our online algorithm on the image separation tasks as reported in the main text.

The results are almost identical to what we obtained in the offline case which confirms the relevance of our algorithm. Interestingly, in the speech separation task the noise is not entirely removed from one of the two voices which explains a "constant" value of the residual in green. However, for the image separation tasks the results are nearly identical.

The learning parameters are presented in the following subsection, and the codes used to produce the respective figures can be found attached to the submission, `NeurIPS_Fig_2B_audio`, `NeurIPS_Fig_2C_image`.

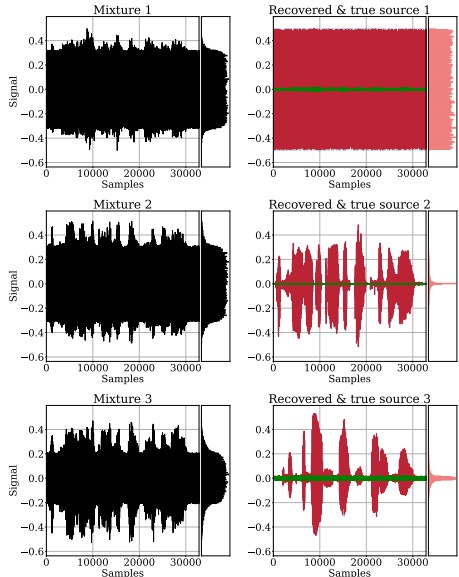

Figure 3: Performance of our online algorithm on the speech separation task. Our algorithm again recovers the sources from mixed signals. As was done for the offline version of our algorithm we show in black the mixed signals and in red (resp. blue) the recovered (resp. true) sources, and in green their difference called Residual. We also show the histogram of the associated distributions.

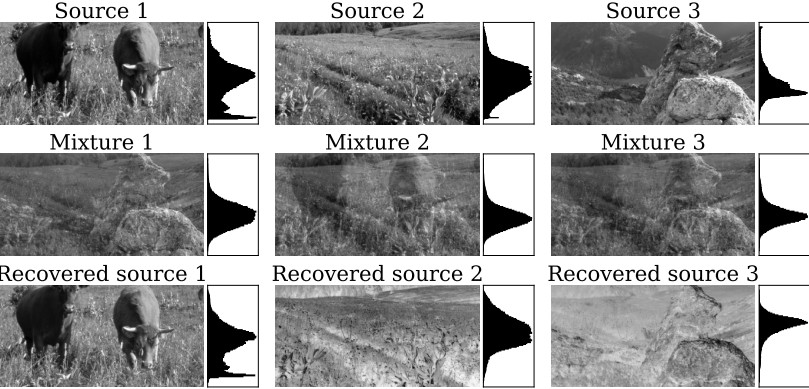

Figure 4: Performance of our online algorithm on the image separation task. Our algorithm again recovers the sources from mixed images. As was done for the offline algorithm we show in each row respectively the original sources, the mixed images, and the recovered alongside their histograms.

## C.3 Experimental details

We implemented both the offline and online version of our similarity-preserving algorithm, Algorithm 1 of the main text. We initialized $\mathbf{W}$ to be a random matrix with i.i.d. mean-zero normal entries with variance 1. We initialized $\mathbf{M}$ to be the identity matrix $\mathbf{I}_d$. We used fixed learning rates for $\eta$ and $\tau$. The use of time-varying learning did not appear to change the results significantly and might instead lead to over-parameter tuning, which is arguably biologically implausible. We nonetheless performed grid-search to find the optimal hyperparameters, we performed a grid search over $\eta_0 \in \{10^{-2}, 10^{-3}, 10^{-4}, 10^{-5}\}$, each multiplied by 1, 2 or 5. For we used $\tau \in \{2, 1.5, 1, 0.9, 0.85, 0.75, 0.5, 0.1\}$. The best performing parameters are reported in Table 1. The results obtained with these parameters are presented in the main text, for Offline SM-ICA Fig.2A,2B and 2C. And the online SM-ICA Fig.3ABC below.

| Algorithm | parameters | synthetic | audio | image |
|---|---|---|---|---|
| **Offline** | $\eta$ , $\tau$ | $5 \cdot 10^{-3}, 0.75$ | $5 \cdot 10^{-4}, 0.85$ | $5 \cdot 10^{-3}, 0.75$ |
| **SM-ICA** | $\mathbf{\Lambda}$ | $[1.0, 1.5, 1.8, 6.07]$ | $[1.8, 6.15, 19.7]$ | $[2.48, 3.06, 6.64]$ |
| **Online** | $\eta$ , $\tau$ | $2 \cdot 10^{-5}, 1.5$ | $2 \cdot 10^{-5}, 1.5$ | $2 \cdot 10^{-5}, 1.5$ |
| **SM-ICA** | $\mathbf{\Lambda}$ | $[1.0, 1.5, 1.8, 6.07]$ | $[1.8, 6.15, 19.7]$ | $[2.48, 3.06, 6.64]$ |

Table 1: Training parameters for comparing our model.

# D Comparison to other models on the synthetic dataset

In this section, we compare numerically the performance of our algorithm against competing models on four different scenarios.

## D.1 Experimental details

**Metrics.** First of all we define the metric used to quantitatively compare the ICA algorithms. As defined in Sec. 2 of the main text, the task of ICA algorithms is to extract the source signals up to a permutation and sign-flipping (Eq. (2)). Therefore, all results are measured by considering all possible pairings of predicted signals and source signals, and measuring the mean-squared error $\varepsilon_{\mathrm{MSE}}$ defined as

$$\varepsilon_{\mathrm{MSE}}(t) = \min_{\mathbf{\Xi}, \mathbf{P}} \frac{1}{td} \sum_{t'=1}^{t} \|\mathbf{s}_{t'} - \mathbf{\Xi}\mathbf{P}\mathbf{y}_{t'}\|^2 \tag{S.10}$$

**Dataset.** We evaluate our algorithm on a synthetic dataset generated by independent and identically distributed samples. The data are generated from meaningful signals, i.e., square-periodic, sine-wave, saw-tooth, and Laplace random noise. The data were chosen with the purpose of including both super- and sub-Gaussian distribution known respectively as leptokurtic ("spiky", e.g., the Laplace distribution) and platykurtic ("flat-topped", the three other source signal).

**Scenarios.** We designed four scenarios: scenario 1, data are white and composed of 3 independent sub-Gaussian sources; scenario 2, data are colored and again composed of only 3 sub-Gaussian sources; scenario 3, data are white and composed of 2 sub-Gaussian sources and 1 super-Gaussian source; scenario 4, Data are colored and composed of 2 sub-Gaussian sources and 1 super-Gaussian source.

**Competing algorithms.** To quantitatively measure the performance we present in Appendix D a numerical comparison of our algorithm with competing algorithms, namely, Herault Jutten algorithm [7], EASI algorithm [9, 5], Bell and Sejnowski's algorithm [3], the Amari algorithm [1], and finally nonlinear Oja algorithm [8, 15], on the synthetic data set, on three different scenarios. These models were designed with NNs in mind and are seminal works on neural ICA algorithms. However, like nonlinear Oja algorithm, which we mentioned in Section 5.3.1, these models suffer from biological implausibility.

| Algo | param. | Scenario 1 | Scenario 2 | Scenario 3 | Scenario 4 |
|---|---|---|---|---|---|
| **Online** | $\eta$ , $\tau$ | $5 \cdot 10^{-3}, 0.75$ | $5 \cdot 10^{-4}, 0.85$ | $5 \cdot 10^{-3}, 0.75$ | $5 \cdot 10^{-4}, 0.85$ |
| **SM-ICA (ours)** | $\Lambda^{-1}$ | $[1.0, 1.5, 1.8]$ | $[1.0, 1.5, 1.8]$ | $[1.0, 1.5, 6.07]$ | $[1.0, 1.5, 6.07]$ |
| **HJ** [7] | $\eta$ | $10^{-4}$ | $10^{-4}$ | $10^{-4}$ | $10^{-4}$ |
| **EASI** [5] | $\eta$ | $10^{-4}$ | $10^{-4}$ | $10^{-4}$ | $10^{-4}$ |
| **Nonlin. Oja** [15] | $\eta$ | $10^{-3}$ | $10^{-3}$ | $10^{-3}$ | $10^{-3}$ |
| **Infomax**[3] | $\eta$ | $5 \cdot 10^{-4}$ | $5 \cdot 10^{-4}$ | $5 \cdot 10^{-4}$ | $5 \cdot 10^{-4}$ |
| **Amari** [1] | $\eta$ | $5 \cdot 10^{-4}$ | $5 \cdot 10^{-4}$ | $5 \cdot 10^{-4}$ | $5 \cdot 10^{-4}$ |

Table 2: Training parameters of the different models on the four different scenarios.

## D.2    Results

We show in Fig. **5a** the comparison between our model and competing algorithm on Scenario 1. Scenario 1 consists of a mixture of sub-Gaussian distribution, which has been pre-whitened. Naturally, in scenario 1, all models perform the task perfectly, with Nonlinear Oja and Amari algorithm being the fastest, our model also performs competitively.

We show in Fig. **5b** the comparison between our model and competing algorithm on Scenario 2. Scenario 2 consists of a mixture of sub-Gaussian distribution, which has not been pre-whitened. In scenario 2, our algorithm is more competitive as it does not require pre-whitening, unlike the algorithms that fail at the task.

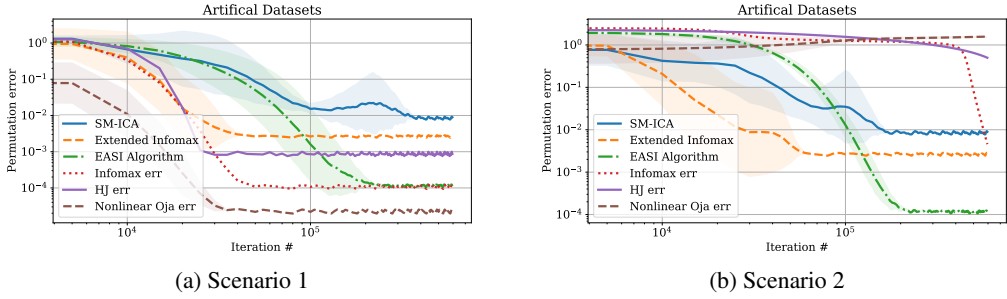

(a) Scenario 1                           (b) Scenario 2

Figure 5: Comparison of the models on mixture of sub-Gaussian distribution, with (a) whitened data, and (b) colored data.

We show in Fig. **6a** the comparison between our model and competing algorithm on Scenario 3. Scenario 3 consists of a mixture of both sub-Gaussian and super-Gaussian distribution, which has been pre-whitened. Again, our algorithm is more competitive as it has the ability to separate sources from a different sign of kurtosis, as explained in the main body of the paper.

We show in Fig. **6b** the comparison between our model and competing algorithm on Scenario 4. Scenario 4 consists of a mixture of both sub-Gaussian and super-Gaussian distributions, which have not been pre-whitened. In this scenario, our algorithm performs one of the best as it does not require pre-whitening and can separate sources with a different sign of kurtosis.

In brief, our model outperforms either outperforms or is competitive with other models on scenarios 2-3-4. Out of the other algorithms, EASI performs well on scenarios 1-2 but fails on tasks 3-4, where data are composed of both sub-Gaussian and super-Gaussian distributions. The other models are known to require pre-whitening and fail on scenarios 2 and 4, naturally. We have prepared a log-log plot of convergence with error bars that illustrate the results mentioned above. These results also confirm that our algorithm performs perfect separation with both whitened and colored data and simultaneously with both sub and super-Gaussian sources.

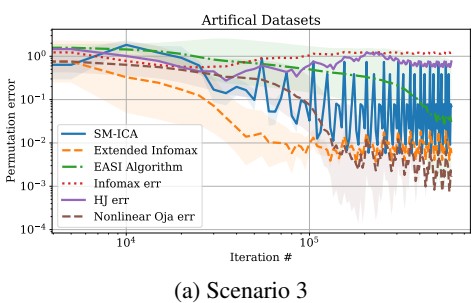

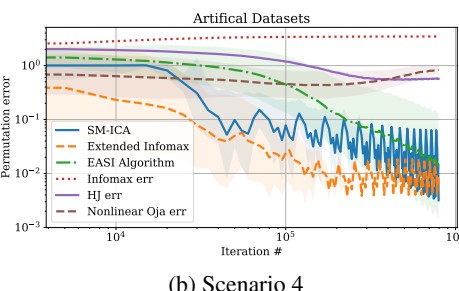

|(a) Scenario 3|(b) Scenario 4|

Figure 6: Comparison of the models on mixture of both sub- and super-Gaussian distribution, with (a) whitened data, and (b) colored data.

# E    Contrasting reconstruction and similarity-preservation NNs

We mention in Section 2 and Section 5 of the paper that neural networks trained with Oja's rule lack biological plausibility, unlike NNs resulting from the similarity matching framework. This statement can be unclear for people not too familiar with the approach. For that purpose, we recall the critical distinction between the similarity matching framework and the reconstruction-based approach from which Oja's results. A more detailed description can be found in [18, 2].

To introduce our notation, the input to the NN is a set of vectors, $\mathbf{x}_t \in \mathbb{R}^n, t = 1, \ldots, T$, with components represented by the activity of $n$ upstream neurons at time, $t$. In response, the NN outputs an activity vector, $\mathbf{y}_t \in \mathbb{R}^m, t = 1, \ldots, T$, where $m$ is the number of output neurons.

**Reconstructive Approach:**    The reconstruction approach starts with minimizing the squared reconstruction error:

$$\min_{\mathbf{W}, \mathbf{y}_{t=1\ldots T} \in \mathbb{R}^m} \sum_t ||\mathbf{x}_t - \mathbf{W}\mathbf{y}_t||^2 = \min_{\mathbf{W}, \mathbf{y}_{t=1\ldots T} \in \mathbb{R}^m} \sum_{t=1}^T \left[ ||\mathbf{x}_t||^2 - 2\mathbf{x}_t^\top \mathbf{W}\mathbf{y}_t + \mathbf{y}_t^\top \mathbf{W}^\top \mathbf{W}\mathbf{y}_t \right]. \tag{S.11}$$

This objective is optimized offline by a projection onto the principal subspace of the input data.

In an online setting, the objective can be optimized by alternating minimization [16]. After the arrival of data sample, $\mathbf{x}_t$: firstly, the objective (S.11) is minimized with respect to the output, $\mathbf{y}_t$, while the weights, $\mathbf{W}$, are kept fixed, secondly, the weights are updated according to the following learning rule derived by a gradient descent with respect to $\mathbf{W}$ for fixed $\mathbf{y}_t$:

$$\dot{\mathbf{y}}_t = \mathbf{W}_{t-1}^\top \mathbf{x}_t - \mathbf{W}_{t-1}^\top \mathbf{W}_{t-1}\mathbf{y}_t, \qquad \mathbf{W}_t \longleftarrow \mathbf{W}_{t-1} + \eta \left( \mathbf{x}_t - \mathbf{W}_{t-1}\mathbf{y}_t \right) \mathbf{y}_t^\top, \tag{S.12}$$

In the NN implementations of the algorithm (S.12), the elements of matrix $\mathbf{W}$ are represented by synaptic weights and principal components by the activities of output neurons $y_j$, Fig. 7a [14].

However, implementing update (S.12)right in the single-layer NN architecture, Fig. 7a, requires non-local learning rules making it biologically implausible. Indeed, the last term in (S.12)right implies that updating the weight of a synapse requires the knowledge of output activities of all other neurons which are not available to the synapse. Moreover, the matrix of lateral connection weights, $-\mathbf{W}_{t-1}^\top \mathbf{W}_{t-1}$, in the last term of (S.12)left is computed as a Gramian of feedforward weights; a non-local operation. This problem is not limited to PCA and arises in nonlinear NNs as well [16, 10].

**Similarity Matching Approach:**    To address these difficulties, [19] derived NNs from similarity-preserving objectives, as presented in Section 3. Such objectives require that similar input pairs, $\mathbf{x}_t$ and $\mathbf{x}_{t'}$, evoke similar output pairs, $\mathbf{y}_t$ and $\mathbf{y}_{t'}$. If the similarity of a pair of vectors is quantified by their scalar product, one such objective is similarity matching (SM):

$$\min_{\forall t \in \{1, \ldots, T\}: \mathbf{y}_t \in \mathbb{R}^m} \frac{1}{2} \sum_{t, t'=1}^T \left( \mathbf{x}_t \cdot \mathbf{x}_{t'} - \mathbf{y}_t \cdot \mathbf{y}_{t'} \right)^2. \tag{S.13}$$

This offline optimization problem is also solved by projecting the input data onto the principal subspace [21, 6, 11]. Remarkably, the optimization problem (S.13) can be converted algebraically to a tractable form by introducing variables $\mathbf{W}$ and $\mathbf{M}$ [20]:

$$\min_{\{\mathbf{y}_t \in \mathbb{R}^m\}_{t=1}^T} \min_{\mathbf{W} \in \mathbb{R}^{n \times m}} \max_{\mathbf{M} \in \mathbb{R}^{m \times m}} [\sum_{t=1}^T (-2\mathbf{x}_t^\top \mathbf{W} \mathbf{y}_t + \mathbf{y}_t^\top \mathbf{M} \mathbf{y}_t) + T \operatorname{Tr}(\mathbf{W}^\top \mathbf{W}) - \frac{T}{2} \operatorname{Tr}(\mathbf{M}^\top \mathbf{M})].$$
(S.14)

In the online setting, first, we minimize (S.14) with respect to the output variables, $\mathbf{y}_t$, by gradient descent while keeping $\mathbf{W}, \mathbf{M}$ fixed [19]:

$$\dot{\mathbf{y}}_t = \mathbf{W}^\top \mathbf{x}_t - \mathbf{M} \mathbf{y}_t. \tag{S.15}$$

To find $\mathbf{y}_t$ after presenting the corresponding input, $\mathbf{x}_t$, (S.15) is iterated until convergence. After the convergence of $\mathbf{y}_t$, we update $\mathbf{W}$ and $\mathbf{M}$ by gradient descent and gradient ascent respectively [19]:

$$W_{ij} \leftarrow W_{ij} + \eta \left( x_i y_j - W_{ij} \right), \qquad M_{ij} \leftarrow M_{ij} + \eta \left( y_i y_j - M_{ij} \right). \tag{S.16}$$

Algorithm (S.15), (S.16) can be implemented by a biologically plausible NN, Fig. 7b. As before, activity (firing rate) of the upstream neurons encodes input variables, $\mathbf{x}_t$. Output variables, $\mathbf{y}_t$, are computed by the dynamics of activity (S.15) in a single layer of neurons. The elements of matrices $\mathbf{W}$ and $\mathbf{M}$ are represented by the weights of synapses in feedforward and lateral connections respectively. The learning rules (S.16) are local, i.e. the weight update, $\Delta W_{ij}$, for the synapse between $i^{\text{th}}$ input neuron and $j^{\text{th}}$ output neuron depends only on the activities, $x_i$, of $i^{\text{th}}$ input neuron and, $y_j$, of $j^{\text{th}}$ output neuron, and the synaptic weight. Learning rules (S.16) for synaptic weights $\mathbf{W}$ and $-\mathbf{M}$ (here minus indicates inhibitory synapses, see Eq.(S.15)) are Hebbian and anti-Hebbian respectively.

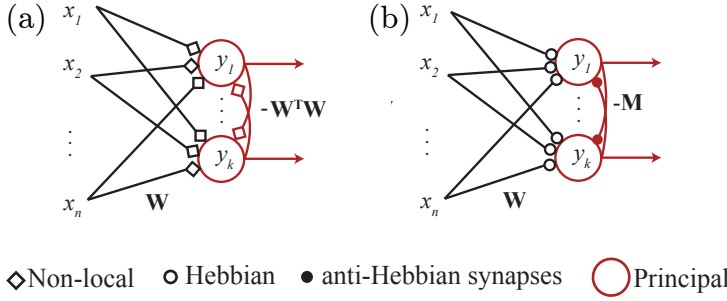

Figure 7: Single-layer NNs performing online (a) reconstruction error minimization (S.11) [14, 16], (b) similarity matching (SM) (S.13) [19].

**Comparison between the two approaches:** We now compare the objective functions of the two approaches. After dropping invariant terms, the reconstructive objective function has the following interactions among input and output variables: $-2\mathbf{x}_t^\top \mathbf{W} \mathbf{y}_t + \mathbf{y}_t^\top \mathbf{W}^\top \mathbf{W} \mathbf{y}_t$ (Eq S.11). The SM approach leads to $-2\mathbf{x}_t^\top \mathbf{W} \mathbf{y}_t + \mathbf{y}_t^\top \mathbf{M} \mathbf{y}_t$, ( Eq S.14). The term linear in $\mathbf{y}_t$, a cross-term between inputs and outputs, $-2\mathbf{x}_t^\top \mathbf{W} \mathbf{y}_t$, is common in both approaches and is responsible for projecting the data onto the principal subspace via the feedforward connections in Fig.1ab. The terms quadratic in $\mathbf{y}_t$'s decorrelate different output channels via a competition implemented by the lateral connections in Fig.1ab and are different in the two approaches. In particular, the inhibitory interaction between neuronal activities $y_j$ in the reconstruction approach depends upon $\mathbf{W}^\top \mathbf{W}$, which is tied to trained $\mathbf{W}$ in a non-local way. In contrast, in the SM approach the inhibitory interaction matrix $\mathbf{M}$ is learned for $y_j$'s via a local anti-Hebbian rule.