# OpenReview forum: "A Normative and Biologically Plausible Algorithm for Independent Component Analysis"
_NeurIPS.cc/2021/Conference — NeurIPS 2021 Spotlight_

### Official Review · Reviewer_h9nv · 2021-07-12

**Rating:** 8
**Confidence:** 5

**Summary:**

The authors propose a biologically plausible neural network implementation for online ICA. It is based on a similarily objective with inspiration from the FOBI procedure. The authors derive an online learning rule and interpret the terms in the updates in a synaptically local manner with the addition of a global activity level signaling mechanism. The authors then validate the method in synthetic, speech, and image datasets.

**Limitations And Societal Impact:**

1. One algorithmic mismatch is the desire to have “online” be a requirement for biologically plausibility, but then time-shuffling the training examples to remove temporal correlations (if I’m understanding Appendix B correctly). Ideally, an online biologically inspired learning algorithm should be applicable to streaming data which may have temporal correlations. Achieving both an online + synaptically local network and learning from correlated timeseries may be beyond the scope of the paper, but it should be addressed at least in the Discussion as a limitation/future direction.

**Main Review:**

Overall, the paper is clearly written, the derivations are sound as far as I can tell, and the results show the applicability to synthetic and real datasets. The results are of interest to readers at the intersection of machine learning and computational neuroscience and possibly hardware implementations of neural networks.

Although I have not run it, the provided code looks like it will reproduce the results from the paper.

Most of my comments are related to clarifying certain points in the paper.

- For the references on lines 111-114, it would be helpful to state whether or not the application of Similarity Matching lead to local or online learning rules for any of those models. Are any of the particular choices that lead to a local, online algorithm for BSS generally applicable? Those references share the most conceptual overlap with this manuscript and it would help to have some clarificaltion on how much technical overlap there is across methods/models.
- Since $||y_t||$ is not clearly a local variable, it requires some interpretation in terms of a potential global modulatory component, which the authors provide. One point that could use some discussion is about how to match timescales since $||y_t||$, at least in the current presentation, needs to modulate at the data sampling rate. Does this point towards an interpretation of faster modulating processes from lines 214-219 being a better potential biological counterpart?
- The difficulty of using Oja’s rule for FOBI on lines 95-97 could be more clear, particularly the sentence that starts on line 95 (what does "true output" mean?). Is another way of stating the problem that Oja’s rule could be applied to $z$ after it is computed, but the goal is to learn a $W$ that applies to $h$?
- Are the results presented in the main text for the offline or online version of the algorithm?

- Related to limitation 1 below, I would have expected the online algorithms to have some amount of “burn-in” time where the errors were higher before they become adapted to the statistics of the dataset, but this is not apparent in the figures. Is this masked by the trial shuffling for training and then unshuffling for display? Can you show (in the appendix) how fast the online variant’s error converges toward the offline solution?
- How fast does the iteration in Eq 16 converge?

- In the second paragraph of the introduction, it might make sense to have a sentence about biologically plausible networks that learn other models similar to ICA, for instance refs [46], [48], and
    - https://journals.plos.org/ploscompbiol/article?id=10.1371/journal.pcbi.1007692
    - https://www.sciencedirect.com/science/article/pii/S0896627317304178
    - https://www.jneurosci.org/content/33/13/5475.short
    - https://journals.plos.org/ploscompbiol/article?id=10.1371/journal.pcbi.1002250

Results on data
- For Fig 2 A, I think it would be more clear to group the 4 mixtures on the left and the 4 recovered sources on the right. Alternating them gives the impression that there is some connection between mixture 1 and source 1, mixture 2 and source 2, etc.

Minor comments

- 280: reconstruct -> reconstruction
- Fig 2 caption: second sentence needs correction

**Time Spent Reviewing:**

5

---

> ### Author Response · Authors · 2021-08-10
> **Response to reviewer h9nv**
>
> We thank reviewer h9nv for their positive review and are responding as thoroughly as possible to the questions raised.  Below we address the comments and will incorporate all feedback in the final version.
>
> - Existing similarity matching algorithms and comparison with NICA and BCA.
>
> We will make clear that the original similarity matching algorithm is more than an algorithm but a procedure for deriving algorithms with local learning rules, which has inspired us greatly. We will also make clear in the introduction that there is not yet any local, online algorithm for BSS generally applicable. We mentioned in the discussion section how we plan on extending our models to be more generalizable.
>
> We will expand on the distinction between nonnegative ICA and our generative model for ICA, stating that nonnegative ICA originated from the seminal work of Plumbley [i], who showed that it could be solved in two steps without having to rely on the higher-order statistics of the sources. We will mention that the work of [69] proposed a two-layered approach combining a principal subspace whitening and nonnegative similarity matching and that the follow-up work from [74] made an essential contribution in offering a single-layered version of the model. These models are different from ours since they use a nonlinear activation function for their output neurons and would not operate when the sources are not nonnegative and do not rely on the higher-order statistics of the data.
>
> Bounded component analysis (BCA) is less well-known than other BSS models but has proved able to separate natural image datasets. There are two crucial distinctions between the model of bounded component analysis and ours. First, they use a clipping activation function when we use a linear activation function, and second, they have a gain update, which they suggest has an interesting interpretation in terms of homeostatic gain adjustment behavior in biological neurons [ii].
>
> When comparing canonical correlation analysis and slow feature analysis, we mentioned in lines 308-312 their connection to second-order blind identification, another particular case of BSS. We also note that we wish to investigate how combining temporal and higher-order statistics would lead to a more general type of BSS in future work. In the final manuscript, when mentioning these models in Section 2.2, we will refer to the discussion section.
>
> [i] Mark D Plumbley, “Algorithms for nonnegative independent component analysis,” IEEE Transactions on Neural Networks, vol. 14, no. 3, pp. 534–543, 2003.
> [ii] Gina G Turrigiano, “Homeostatic plasticity in neuronal networks: the more things change, the more they stay the same,” Trends in neurosciences, vol. 22, no. 5, pp. 221–227, 1999.
>
> - Regarding the time-scale of the modulating factor $||y_t||^2$
>
> The reviewer is correct in pointing out the requirement on fast $||y_t||$ signaling. From the time scale perspective, the most likely mechanisms are the local field potential and calcium which can act on synaptic plasticity time scales. We will rewrite the paragraph to indicate the time scale associated with various suggested mechanisms.
>
> - Clarifying the difficulty of using Oja's rule for FOBI:
>
> Yes, that is exactly what we meant. We will refer more clearly to the variables used for the learning and those that estimate the sources.
>
> - Results presented in the main text:
>
> We realized we added the offline instead of online the main text. We showed both offline and online results in the appendix. In the revised version we will swap the offline and online in the main text.
>
> - Convergence and learning:
>
> There is indeed a “burn-in” period for convergence of the algorithm. We have prepared both a numerical comparison to other models and a quantitative measure of the accuracy of our models. These results show how fast the residual (shown in green in the figures) tends to zero as a function of the number of iterations.
>
> The shuffling is not necessary; our model works even without it. However, we wanted to ensure that the reader might not think that we use temporal correlation instead of higher-order statistics to separate the sources. We will emphasize that in the revised version of the manuscript.
>
> - Convergence of the dynamic Eq.16:
>
> It converges rather fast, it usually takes 5-6 iterations for convergence, depending on the tolerance, (we set it at 10^-3).
>
> - References mentioned by the reviewer:
>
> We will add the references to the work mentioned by the reviewer and will emphasize the importance of spiking models.
>
> - Regarding Fig 2.A
>
> We appreciate the feedback on the arrangement of the figures in Fig 2A and have the mixtures at the top and recovered sources at the bottom to avoid confusion. With the extra space, we will also try to increase the font size. We will also mention in the main text that we have a zoomed-in version in the appendix.
>
> - Minor comments:
>
> Thank you also for the minor comments, we will make changes accordingly.

---

> > ### Comment · Reviewer_h9nv · 2021-08-14
> > **Author's response addresses my comments.**
> >
> > This addresses all of my comments. These clarifications along with the changes from other reviews should strengthen the paper.

---

### Official Review · Reviewer_QCCE · 2021-07-13

**Rating:** 8
**Confidence:** 4

**Summary:**

This work proposes a novel similarity-preserving objective function for targeting ICA. They utilize this objective function to derive a gradient descent-based learning algorithm that takes advantage of this objective function. This algorithm is mapped to a single-layer neural network that is able to separate independent sources without pre-processing. They compare the proposed learning algorithms with oja-based methods, BCM, and Modulated Hebbian learning.

**Limitations And Societal Impact:**

The weaknesses and limitations of the work are discussed in the main review. The work would have any obvious direct negative societal impacts.

**Main Review:**

The strengths of this work are summarized as follows:
- The work proposes a novel perspective on a topic that has not been heavily explored.
- The development of biologically plausible neural networks could provide valuable insight into the brain’s computational properties. In addition, a biologically plausible method of ICA is particularly valuable.
- The theoretically grounded algorithm successfully integrates insights from the FOBI procedures into a neural network structure. The experimental results indicate that the implemented neural network can successfully perform ICA.

However, the observed weaknesses of the paper are summarized as:
- The experimental results presented demonstrates that the methodology can recover the original sources from a Mixed signal. But it is difficult to quantify the degree of that success for comparison with other methodologies. The authors should include a quantitative evaluation of the algorithm's effectiveness and quality of the recovered signals.
- While the work contrasts the proposed work with the Oja-based, BCM and Modulated Hebbian methodologies, the work does not present experimental comparisons to highlight the benefits of the proposed approach. The authors should highlight these distinctions.
- The authors utilize the proposed scheme only in a single layer architecture. The authors should demonstrate the scalability of the proposed scheme using a deeper model.

In addition to the above comments, we hope the following can help the authors further improve their work.
- The font size of some text in both figures should be increased.
- While computational restrictions are not necessarily critical in the target scenario, it would be good to provide such information to evaluate the algorithm’s overhead properly.
- Since the two-compartment neural model is uncommon, it would be good to include a concrete mathematical model of its functionality.
- The authors make the claim that a minimal biological neuron model must adhere to the online learning and local update rule constraints. It would be good for the authors to motivate this assumption as it is not self-evident.


**Time Spent Reviewing:**

5

---

> ### Author Response · Authors · 2021-08-10
> **Response to reviewer QCCE**
>
> We thank reviewer QCCE for their positive review and are responding as thoroughly as possible to the questions raised. Below we address the comments and will incorporate all feedback in the final version.
>
> ## - Regarding the numerical evaluations of our model:
>
>
> Thank you for raising this point. Yes, we have evaluated our model against other ICA algorithms. Using the extra page and the appendix section, we will add these results to the final manuscript.
>
> - Evaluation Scenarios:
>
> We evaluated the models on four different scenarios.
>
> Scenario 1: Data are white and composed of 3 independent sub-Gaussian sources
> Scenario 2: Data are colored and again composed of only 3 sub-Gaussian sources.
> Scenario 3: Data are white and composed of 2 sub-Gaussian sources and 1 super-Gaussian source.
> Scenario 4: Data are colored and composed of 2 sub-Gaussian sources and 1 super-Gaussian source.
>
> - Metric:
>
> To quantify the performance of the algorithms, we use the mean-squared error,
> $$ error(t) = \frac{1}{td} \sum_{t=1}^T \| x_t - Py_t \|^2; $$
> With $t$ the number of samples and $d$ the dimension of the sources.
>
> - Competing algorithms:
>
> The algorithms that we evaluated against are the following:
> The Herault Jutten (HJ) algorithm [i]:
> The unmixing matrix is sought in the form $W = ( I+ S^{-1})$ and the off-diagonal elements of S are updated using the rule:
> $$ S_{t+1} = S_{t} + \eta g(y_t) h(y_t)^\top ~~, $$
> with $h$ and $g$ two possibly different linear or nonlinear component-wise functions. For sub-Gaussian distributions we use $g(y) = y^3$ and $h(y) = y$.
>
> The EASI algorithm [ii]-[iii] is closely related to a nonlinear PCA-type network.
> $$ W_{t+1} = W_{t} - \eta ( y_t y_t^\top - I + g(y_t) h(y_t)^\top - h(y_t) g(y_t)^\top ) W_{t} ~~. $$
> For sub-Gaussian distributions we use $g(y) = y$ and $h(y) = tang(y)$.
>
> The Bell and Sejnowski algorithm [iv] from an information theoretic approach (Infomax):
> $$ W_{t+1} = W_{t} + \eta ( W_{t}^{-1} + g(y_t) x_t^\top ) ~~. $$
> We use $g(y) = -2 tanh(y)$.
>
> The Amari algorithm [v], based on Infomax above.
> $$ W_{t+1} = W_{t} + \eta ( I - 2 g(y_t) y_t^\top ) W_{t} $$
> We use $g(y) = -2 tanh(y)$.
>
> And finally the nonlinear Oja algorithm [vi]-[vii]:
> $$ W_{t+1} = W_{t} + \eta ( g(y_t) x_t^\top - g(y_t) g(y_t)^\top W_t^\top) $$
> with again $g(y) = tanh(y)$.
>
> We chose these models because they were designed with neural architectures in mind and are seminal works on neural ICA algorithms. However, like Nonlinear Oja that we mentioned in Section 5.3.1, these models suffer from biological implausibility. We will describe their limitations in the appendix and leave it for the interested reader.
>
> - Results:
>
> In brief, our model outperforms others models on scenarios 2-3-4 and is competitive with the Amari algorithm. The Amari model, an improved version of Infomax, is known to be significantly faster. Naturally, in scenario 1, all models perform the task perfectly, with the Nonlinear Oja and Amari algorithm being the fastest.
> Out of the other algorithms, EASI performs well on scenarios 1-2 but fails on tasks 3-4, where data are composed of both sub-Gaussian and super-Gaussian distributions. The other models are known to require pre-whitening and fail on scenarios 2 and 4, naturally.
> We have prepared a log-log plot of convergence with error bars that illustrate the above mentioned results.
>
> The results show that our algorithm performs perfect separation with both whitened and colored data, and with sources both sub and super-Gaussian simultaneously.
>
>
> [i] Jutten, Christian, and Jeanny Herault. "Blind separation of sources, part I: An adaptive algorithm based on neuromimetic architecture." Signal processing 24.1 (1991): 1-10.
> [ii] Laheld, Beate, and Jean-François Cardoso. "Adaptive source separation with uniform performance." Proc. EUSIPCO. Vol. 1. 1994.
> [iii] Cardoso, J-F., and Beate H. Laheld. "Equivariant adaptive source separation." IEEE Transactions on signal processing 44.12 (1996): 3017-3030
> [iv] Bell, A. J., & Sejnowski, T. J. (1995). An information-maximization approach to blind separation and blind deconvolution. Neural computation, 7(6), 1129-1159.
> [v] Amari, Shun-ichi, Andrzej Cichocki, and Howard Hua Yang. "A new learning algorithm for blind signal separation." Advances in neural information processing systems. Morgan Kaufmann Publishers, 1996.
> [vi] Karhunen, Juha, Liuyue Wang, and Ricardo Vigario. "Nonlinear PCA type approaches for source separation and independent component analysis." Proceedings of ICNN'95-International Conference on Neural Networks. Vol. 2. IEEE, 1995.
> [vii] Oja, E. (1997). The nonlinear PCA learning rule in independent component analysis. Neurocomputing, 17(1), 25-45.
>
>
>
> ## - Regarding the scalability and stacking of our model:
>
> ICA is inherently a linear mixture of sources, and, as a result, the idea of stacking such models might not necessarily be useful. It would be interesting to consider a “convolutional” version of this network, but it is not clear what benefits that would bring. We will mention the ongoing work on nonlinear ICA [viii] - [ix], which is of great interest to us, and is a perfect candidate for multi-layered architectures. Before stacking models like the one, we proposed we would require the introduction of nonlinearities, which is the subject of ongoing research.
>
> [viii] Hyvarinen, A., Sasaki, H., & Turner, R. (2019, April). Nonlinear ICA using auxiliary variables and generalized contrastive learning. In The 22nd International Conference on Artificial Intelligence and Statistics (pp. 859-868). PMLR.
> [ix] Khemakhem, I., Kingma, D., Monti, R., & Hyvarinen, A. (2020, June). Variational autoencoders and nonlinear ica: A unifying framework. In International Conference on Artificial Intelligence and Statistics (pp. 2207-2217). PMLR.
>
>
> ## - Regarding the supplementary comments:
>
> Thank you for the comment, we will increase the font size and make the figures more readable using the extra page in the final manuscript, computational and memory complexity of the models.
>
> ### - Regarding the two-compartment model of neurons used.
>
> We will add more context to our statement and refer to past and recent work on the subject. We will mention in the appendix how such neurons, e.g Pyramidal cells, consist of two compartments: a dendritic compartment, as well as a somatic compartment with two distinct membrane potentials [x]. The modeling of principal neurons using a two-compartment formalism has been validated by several recent studies [xi]-[xii]-[xiii].
>
>    As the reviewer rightly said, discussion of two-component neurons is less prevalent in the machine learning literature. We will mention the recent work from [xiv]-[xv]-[xvi] that certainly sparked a new interest in incorporating dendritic structures to bio-inspired machine learning approaches. We will also refer the reader to the new review from [viii].
>
> ### - Regarding the constraints of online learning and locality of the learning rules.
>
> We will refer the reader to important works in the field [xvii]-[xviii]-[xix] as well as the fundamental concepts of Hebbian learning and the follow-up from Oja work Linsker and others [xx]-[xxi]-[xxii], on online local learning algorithms for biologically plausible neural networks.
>
> We will clarify that biologically plausible algorithms must be formulated in the online (or streaming), rather than offline (or batch), setting. This means that input data are streamed to the algorithm sequentially, one sample at a time, and the corresponding output must be computed before the next input sample arrives. The output communicated to downstream neurons cannot be modified in the future. Also, a neuron cannot store individual past inputs or outputs except in a highly compressed format limited to synaptic weights and a few state variables.
>
> In biologically plausible neural networks, learning rules must be local. This means that the synaptic weight update may depend on the information of only the two neurons a synapse connects, as for example, in Hebbian learning. Precise activities of other neurons are not physically available to a synapse and therefore including them into learning rules would be biologically implausible.
>
> [x] Paul F Pinsky and John Rinzel. Intrinsic and network rhythmogenesis in a reduced traub model for CA3 neurons. Journal of Computational Neuroscience , 1(1-2):39–60, 1994.
>
> [xi] Gasparini, Sonia, and Jeffrey C. Magee. "State-dependent dendritic computation in hippocampal CA1 pyramidal neurons." Journal of Neuroscience 26.7 (2006): 2088-2100.
> [xii] Nelson Spruston. Pyramidal neurons: dendritic structure and synaptic integration. Nature Reviews Neuroscience, 9(3):206–221, 2008.
> [xiii] Katz, Y., Menon, V., Nicholson, D. A., Geinisman, Y., Kath, W. L., & Spruston, N. (2009). Synapse distribution suggests a two-stage model of dendritic integration in CA1 pyramidal neurons. Neuron, 63(2), 171-177.
>
> [xiv] Guerguiev, Jordan, Timothy P. Lillicrap, and Blake A. Richards. "Towards deep learning with segregated dendrites." Elife 6 (2017): e22901.
> [xv]Sacramento, J., Ponte Costa, R., Bengio, Y., & Senn, W. (2018). Dendritic cortical microcircuits approximate the backpropagation algorithm. Advances in Neural Information Processing Systems, 31, 8721-8732.
> [xvi] Richards, Blake A., and Timothy P. Lillicrap. "Dendritic solutions to the credit assignment problem." Current opinion in neurobiology 54 (2019): 28-36.
> [xvii] Chavlis, Spyridon, and Panayiota Poirazi. "Drawing inspiration from biological dendrites to empower artificial neural networks." Current Opinion in Neurobiology 70 (2021): 1-10.

---

> > ### Author Response · Authors · 2021-08-10
> > **References**
> >
> > [xviii] D. Hebb. The organization of behavior: A neurophychological study. Wiley Interscience, New York, 1949.
> > [xix] K. Fukushima. Neocognitron: A self-organizing neural network model for a mechanism of pattern recognition unaffected by shift in position. Biological cybernetics, 36(4):193–202, 1980.
> > [xx] J. Hopfield. Neural networks and physical systems with emergent collective computational abilities. Proceedings of the National Academy of Sciences of the United States of America, 79(8):2554–2558, 1982.
> >
> > [xxi] J. Hertz, A. Krogh and R. G. Palmer (1991) Introduction to the Theory of Neural Computation. Addison-Wesley, Redwood City CA.
> > [xxii] S. Haykin (1994) Neural networks. Prentice Hall, Upper Saddle River, NJ.
> > [xxiii] Gerstner, W., Kistler, W. M., Naud, R., & Paninski, L. (2014). Neuronal dynamics: From single neurons to networks and models of cognition. Cambridge University Press.

---

> > > ### Comment · Reviewer_QCCE · 2021-08-16
> > > **Response**
> > >
> > > Thank you for the response. The authors have agreed to incorporate additional evaluations as well as clarify some points of interest. I thank the authors for their effort in addressing these comments. With this, they have addressed all of my comments.

---

> > > > ### Author Response · Authors · 2021-08-17
> > > > **Response**
> > > >
> > > > Thank you for reading and replying to our response. We really appreciated the feedback and your time.

---

### Official Review · Reviewer_up9Q · 2021-07-14

**Rating:** 7
**Confidence:** 3

**Summary:**

This paper presents a normative study by combining the inspirations from fourth-order blind identification and similarity matching to derive a novel learning rule to achieve independent component analysis in a single layer recurrent network. The derived learning rule only depends on the overall neuronal responses in the network, rather than the detailed population activity pattern, which makes the learning rule more biologically plausible compared with previous work. The paper also justifies some possible neural substrates to access the information of overall neuronal activities in a neuronal population.

**Limitations And Societal Impact:**

Yes

**Main Review:**

This work is based on previous work of similarity matching and marries it to fourth-order blind identification to achieve independent component analysis. The derived learning rule is novel, and the mathematical analysis is solid. The whole paper is structure-wise and well written. Overall I think it is a good normative study worth publishing.

Major concerns
* Motivation of minimax optimization.
When I read the paper I feel the flow has a jump at section 4.1 in that I cannot get the motivation of converting the original minimized objective function into a minimax objective function, and the motivation of proposing the particular form of minimized objective. Is this minimax objective a necessary step to implement the algorithm in a network? Or the previous similarity matching algorithm optimize a minimax objective function? I wish the authors could add more motivations in a revised manuscript.


Minor concerns
* Eq. 5: is $\Lambda$ a parameter that needs to be optimized? Reading through the whole paper I don’t see how $\Lambda$ is constrained.

* Eq. 13: I can tell $\alpha_t  = |y_t|^2$ resulted from a rotation matrix defined in step 2b. However, I don’t see why $\alpha_t = |s_t|^2$. Is there an implicit assumption that the mixing matrix A (Eq. 1) is also a rotation matrix?

* line 155: Eq. 10 --> Eq. 8

* Eq. 18: the authors called this a thresholded nonlinearly modulated Hebbian rule. I don’t see how the thresholding process comes from. I know the $\Lambda$ determines LTP and LTD but there seems to be no thresholding like all or none process similar with thresholding in spike generation.

* The Eq. 6 and definition of $\Gamma_x$ at line 140 can be simplified as the result appearing Eq. 12, which could simplifies the math equations a lot.

* Comparing Fig. 1B and neural dynamics shown in Eq. 16, the recurrent connection weight should be $I-M$ instead of $M$ since typically there is a leak term.


**Time Spent Reviewing:**

3

---

> ### Author Response · Authors · 2021-08-10
> **Response to reviewer up9Q**
>
> We thank reviewer up9Q for their positive review and are responding as thoroughly as possible to the questions raised.
>
> ### - Regarding the introduction of the min-max objective function:
>
> We will clarify the essential motivation for the min-max objective function: the gradient descent/ascent optimization of the min-max problem gives us the Hebbian/anti-Hebbian rules right away. The original similarity matching from [66] already led to local learning rules, but obtaining the online optimization and identifying the synaptic matrices was unclear. The follow-up work from most of the authors [70] clarified the procedure for obtaining local learning rules from simple objectives by introducing auxiliary variables and variable substitution tricks. It is then these auxiliary variables that identify with synaptic connections.
> We will expand on the procedure in the main text and add to the appendix how to recover the original objective from the min-max objective function by setting the derivative with respect to $W$ and $M$ to zero.
>
>
> ### - Regarding the constraint on $\Lambda$:
>
> We will make this fact more central. We mentioned in line 121 that $\Lambda^2$ is a diagonal matrix with distinct finite positive entries. This is the only necessary constraint. We have ways to optimize $\Lambda$, but we believe it adds complexity to the model while not being central to the paper.
>
> ### - Regarding the fact that $\alpha_t = || s_t ||^2$:
>
> We will make this fact clearer as it is an essential insight for the understanding of the method. No, we do not assume that A is a rotation matrix, $A$ only has to be full rank.
> The fact that $\alpha_t=  \| s_t \|^2$ results from the fact that, after whitening, i.e., multiplying by $ C^{-1/2}_x $, the original samples are now only a rotation away from the initial sources.
>
>
> We will correct the wrong referencing of Eq.10 instead of Eq.8 in line 155.
>
>
> ### - Regarding the naming of the learning rule:
>
> We chose thresholding as a synonym for a value separating two regimes of plasticity, but it can be confusing within the field. We will drop the “thresholded” to the characterization of the type of learning rules.
>
> ### - Regarding Eq.6 and the introduction of $\Gamma_x$:
>
> We have made multiple attempts to have a simplified version of Eq.6. We thought that it would not be apparent where the $\alpha_t$ term comes from. We will try to identify $\alpha_t$ with $||s_t||^2$ early on, and then identify the term with $|| y_t ||^2$, a known accessible quantity to the network later to simplify the maths.
>
> ### - The discrepancy between the Figure and the matrix of lateral connections $M$:
>
> We agree with the reviewer and will indicate that the M in the diagram only involves the off-diagonal part of M. We will elaborate on these points and the fact that the network can be made autapse-free, i.e., without self-coupling neurons as in [i]-[ii].
>
> [i] Pehlevan, C., Sengupta, A. M., & Chklovskii, D. B. (2017). Why do similarity matching objectives lead to Hebbian/anti-Hebbian networks?. Neural computation, 30(1), 84-124.
> [ii] Minden, V., Pehlevan, C., & Chklovskii, D. B. (2018, October). Biologically plausible online principal component analysis without recurrent neural dynamics. In 2018 52nd Asilomar Conference on Signals, Systems, and Computers (pp. 104-111). IEEE.

---

### Official Review · Reviewer_EXE1 · 2021-07-16

**Rating:** 7
**Confidence:** 3

**Summary:**

The paper proposes a biologically plausible neural network (BPNN) with an independent component analysis (ICA) objective loss function to solve blind source separation problem. The main contribution of this paper lies in that, it extends BPNN for principal component analysis (PCA) [56] to BPNN for ICA with the fourth-order blind identification (FOBI). Simulation experiments demonstrate that the proposed approach is effective in solving blind source separation problem.

**Ethical Concerns:**

No.

**Limitations And Societal Impact:**

Yes.

**Main Review:**

The propose approach is interesting and has innovations both theoretically and algorithmically. The paper is well organized and presented.

One shortage of the paper lies in that the experiment part is not strong. The experiments contain three datasets of a synthetic data, a speech data, and an image data. The results are mostly presented with illustrations rather than quantitative criterions. It also lacks of comparisons with other approaches. Adding quantitative evaluations and comparisons can help demonstrate the advantages of the proposed approach.

It is also interesting to ask what are the possible applications of the proposed approaches? What can we benefit from the biologically plausible neural network in the blind source separation problem? Can the proposed model help explain biological mechanisms in the brain?

============================

Post-rebuttal:

The authors have addressed most of my comments. I have raised my score from 6 to 7.

**Time Spent Reviewing:**

4

---

> ### Author Response · Authors · 2021-08-10
> **Response to reviewer EXE1**
>
> We thank reviewer EXE1 for their positive review and are responding as thoroughly as possible to the questions raised. Below we address the comments and will incorporate all feedback in the final version.
>
> ### - Regarding the numerical evaluations:
>
> Thank you for raising this point. Yes, we have evaluated our model against other ICA algorithms. Using the extra page and the appendix section, we will add these results to the final manuscript.
>
> - Evaluation scenarios: We evaluated the models on four different scenarios.
>
> Scenario 1: Data are white and composed of 3 independent sub-Gaussian sources
> Scenario 2: Data are colored and again composed of only 3 sub-Gaussian sources.
> Scenario 3: Data are white and composed of 2 sub-Gaussian sources and 1 super-Gaussian source.
> Scenario 4: Data are colored and composed of 2 sub-Gaussian sources and 1 super-Gaussian source.
>
> - Metric:
>
> To quantify the performance of the algorithms, we use the mean-squared error,
> $$ error(t) = \frac{1}{td} \sum_{t=1}^T || x_t - Py_t ||^2; $$
> With $t$ the number of samples and $d$ the dimension of the sources.
>
> - Competing algorithms: The algorithms that we evaluated against are the following:
>
> The Herault Jutten (HJ) algorithm [i]:
> The unmixing matrix is sought in the form $W = ( I+ S^{-1})$ and the off-diagonal elements of S are updated using the rule:
> $$ S_{t+1} = S_{t} + \eta g(y_t) h(y_t)^\top ~~, $$
> with $h$ and $g$ two possibly different linear or nonlinear component-wise functions. For sub-Gaussian distributions we use $g(y) = y^3$ and $h(y) = y$.
>
> The EASI algorithm [ii]-[iii] is closely related to a nonlinear PCA-type network.
> $$ W_{t+1} = W_{t} - \eta ( y_t y_t^\top - I + g(y_t) h(y_t)^\top - h(y_t) g(y_t)^\top ) W_{t} ~~. $$
> For sub-Gaussian distributions we use $g(y) = y$ and $h(y) = tang(y)$.
>
> The Bell and Sejnowski algorithm [iv] from an information theoretic approach (Infomax):
> $$ W_{t+1} = W_{t} + \eta ( W_{t}^{-1} + g(y_t) x_t^\top ) ~~. $$
> We use $g(y) = -2 tanh(y)$.
>
> The Amari algorithm [v], based on Infomax above.
> $$ W_{t+1} = W_{t} + \eta ( I - 2 g(y_t) y_t^\top ) W_{t} $$
> We use $g(y) = -2 tanh(y)$.
>
> And finally the nonlinear Oja algorithm [vi]-[vii]:
> $$ W_{t+1} = W_{t} + \eta ( g(y_t) x_t^\top - g(y_t) g(y_t)^\top W_t^\top) $$
> with again $g(y) = tanh(y)$.
>
> We chose these models because they were designed with neural architectures in mind and are seminal works on neural ICA algorithms. However, like Nonlinear Oja that we mentioned in Section 5.3.1, these models suffer from biological implausibility. We will describe their limitations in the appendix and leave it for the interested reader.
>
> - Results:
>
> In brief, our model outperforms others models on scenarios 2-3-4 and is competitive with the Amari algorithm, which is an improved version of Infomax and is known to be significantly faster. Naturally, in scenario 1, all models perform the task perfectly, with the Nonlinear Oja and Amari algorithm being the fastest.
> Out of the other algorithms, EASI performs well on scenarios 1-2 but fails on tasks 3-4, where data are composed of both sub-Gaussian and super-Gaussian distributions. The other models are known to require pre-whitening and fail on scenarios 2 and 4, naturally.
> We have prepared a log-log plot of convergence with error bars that illustrate the above mentioned results.
>
> The results show that our algorithm performs perfect separation with both whitened and colored data, and with sources both sub and super-Gaussian simultaneously.
>
>
> [i] Jutten, Christian, and Jeanny Herault. "Blind separation of sources, part I: An adaptive algorithm based on neuromimetic architecture." Signal processing 24.1 (1991): 1-10.
> [ii] Laheld, Beate, and Jean-François Cardoso. "Adaptive source separation with uniform performance." Proc. EUSIPCO. Vol. 1. 1994.
> [iii] Cardoso, J-F., and Beate H. Laheld. "Equivariant adaptive source separation." IEEE Transactions on signal processing 44.12 (1996): 3017-3030
> [iv] Bell, A. J., & Sejnowski, T. J. (1995). An information-maximization approach to blind separation and blind deconvolution. Neural computation, 7(6), 1129-1159.
> [v] Amari, Shun-ichi, Andrzej Cichocki, and Howard Hua Yang. "A new learning algorithm for blind signal separation." Advances in neural information processing systems. Morgan Kaufmann Publishers, 1996.
> [vi] Karhunen, Juha, Liuyue Wang, and Ricardo Vigario. "Nonlinear PCA type approaches for source separation and independent component analysis." Proceedings of ICNN'95-International Conference on Neural Networks. Vol. 2. IEEE, 1995.
> [vii] Oja, E. (1997). The nonlinear PCA learning rule in independent component analysis. Neurocomputing, 17(1), 25-45.
>
>
> ### - Regarding the possible applications of our approach:
>
> We agree with the reviewer. Besides the standard signal processing applications, we are particularly interested in implementing online algorithms into neuromorphic devices. A first attempt using a competing model (EGHR compared in the paper section 5) has been proposed in [vii].
>
> [vii] Fouda, M. E., Neftci, E., Eltawil, A., & Kurdahi, F. (2018). Independent component analysis using RRAMs. IEEE Transactions on Nanotechnology, 18, 611-615.
>
>
> ### - Finally regarding the other possible benefits of our approach:
>
> These are two exciting questions, and answering is our ultimate goal. We wish that we could have answered those questions within this single contribution. However, what we propose in this paper is a fresh new look at the problem rather than an entirely conclusive and executed research program. We believe that compared to most of the previous work on biologically plausible ICA, this is the first one that proposes a different cost function to build networks with a single layer. We will ask those fundamental motivational questions within the paper and describe where we fall short more precisely.
>
> As mentioned above, we will expand our discussion on possible neural structures that could perform in a manner similar to our proposed network. The research on ICA for explaining neural learning has sadly been very slow in recent years, after seminal exciting works from Bell, Sejnowski, Oja, Hyvarinen, and many others.  While acknowledging that there is so much more to be understood about blind source separation in animal brains, we believe that we picked up where these early works left and made an important advance in the search for a biologically plausible multichannel ICA network.

---

> > ### Comment · Reviewer_EXE1 · 2021-08-31
> > **Post-rebuttal comment**
> >
> > Thanks for the response. The authors have addressed most of my comments. I have raised my score from 6 to 7.

---

### Official Review · Reviewer_yveC · 2021-07-16

**Rating:** 7
**Confidence:** 4

**Summary:**

This is a well-written paper containing novel ideas. The paper is motivated by an interesting question: how might fundamental computations such as independent component analysis (ICA) be performed by brains? The authors develop a neat algorithm to perform ICA. The paper claims that the developed algorithm is biologically plausible, i.e. might run in a similar form in a biological brain.

# Update (2021-08-20)

Following the author's response from 2021-08-20, all my questions are answered. I had misunderstood the semantics of $x_t$. I updated my score.


**Limitations And Societal Impact:**

Yes

**Main Review:**

# Update (2021-08-20)

Following the author's response from 2021-08-20, all my questions are answered. I had misunderstood the semantics of $x_t$. I updated my score.


# Overview

This is a well-written paper containing novel ideas. The paper is motivated by an interesting question: how might fundamental computations such as independent component analysis (ICA) be performed by brains? The authors develop a neat algorithm to perform ICA. The paper claims that the developed algorithm is biologically plausible, i.e. might run in a similar form in a biological brain.

My main criticism of the paper is that unless there is a misunderstanding on my part, the described algorithm may not be biologically plausible. I would be grateful if the authors could respond to my questions Q1, Q2 and Q3 below. If the authors argue successfully in this review process and in the paper that the proposed algorithm is biologically plausible, I will change the review score that I am assigning today.

# Questions for the authors

- Q1. Unless I misunderstood the paper, the algorithm that the authors propose and claim to be biologically plausible does not appear biologically plausible to me. In Algorithm 1 and Section 5.1, the authors explain that computing independent components involves updating the “weight matrix W encoded by the feedforward synapses connecting the input neurons to the output neurons”. This would imply that synapses re-learn completely new weights every time that the set of neurons computes independence components. It would be very surprising if the brain worked in such a manner. To be concrete: while listening to a conference speaker in a full conference room, my auditory system constantly has to separate the speaker’s voice from the background noise. As this computation happens, my auditory system’s synapses do not need to re-wire themselves in every fraction of a second, as I process a new chunk of audio signal. Instead, my auditory system in the brain developed some kind of noise filter when I was an infant, effectively learning synaptic weights that let me separate human voice from background noise across a wide range of environments. These weights may have changed somewhat over the years, but they certainly do not need to update every moment that I am processing auditory information. In essence, the authors are using (synaptic) weights that we know to learn slowly as fast-updating working memory. A biological neuron cannot learn fast enough to support running Algorithm 1 in real-time.

- Q2. It would be useful if the authors could add details on how the proposed algorithm would map to known structures in say the human brain. The authors mention that “Various brain areas, such as the early visual, auditory, and olfactory systems, are known to identify 17 sources from their mixtures effortlessly [20, 85, 5, 52], an unsupervised task also known as blind 18 source separation (BSS) [19].” Since NeurIPS is mainly a machine learning conference, it would be help if the authors could expand on this sentence, survey in more detail what relevant information is known, and describe where they see the proposed algorithm fit in. For say visual processing, would the proposed algorithm run in the neurons of the primary visual cortex (V1)?

- Q3. Have the authors benchmarked against any other ICA implementations?


# Other suggestions and ideas for improvement


- Algorithm 1 contains notation that I find odd: just after “run the following until convergence:”, it looks like we assign to a derivative. Is that a typo? The line should probably be something closer to equation (16).


- The proposed algorithm is very specific. Is there biological evidence suggesting that the brain is running an algorithm like the one proposed?

- Reference [5] cited by the authors suggests that auditory scene analysis might involved not only the bottom-up kind of computing that is present in the proposed algorithm, but also top-down processing: ”In humans, auditory scene analysis involves both bottom-up and top-down processes (reviewed in Bregman, 1990; Carlyon, 2004; Feng & Ratnam, 2000; Näätänen, Tervaniemi, Sussman, Paavilainen, & Winkler, 2001). Bottom-up mechanisms are “stimulus-driven,” meaning that they operate only or primarily on cues present in the acoustic signal itself; they are largely automatic and obligatory, meaning that they do not critically depend on attention (although this is currently a disputed issue). In contrast, top-down processes depend on a listener's prior experience and expectations, and thus involve higher-level cognitive processes, such as learning, memory, and attention.” It would be useful to point out that a bottom-up algorithm like the one described in this paper would likely only constitute one part of a more complex system to perform e.g. signal processing computations.

**Time Spent Reviewing:**

10

---

> ### Author Response · Authors · 2021-08-10
> **Response to reviewer yveC - Question 1 and 2**
>
> We thank reviewer yveC for their review and are responding as thoroughly as possible to the questions raised. We also hope that we will be able to discuss it further with the reviewer in the upcoming discussion period.
>
> ## - Response to Q1
>
> We agree with the reviewer that the traditional exposition of the BSS/ICA problem via the cocktail party problem can be misleading as there are multiple version cocktail-like party problems that the brain might be solving.
>
> One version of the cocktail party problem, as the reviewer mentioned, is the extraction of a source in a non-stationary noisy environment. However, we consider another type of cocktail party-like problem where the task is to separate multiple sources simultaneously in a stationary environment. We will explicitly refer to it as “ source separation in a stationary environment.”
>
> We will also make clear to the reader that mathematically, it is the latter version of the cocktail party problem that ICA can address. In our expanded discussion for the machine learning community, we will explicitly mention neuroscience problems to which BSS/ICA like ideas may be applied. We will discuss, among others, the seminal work of  Bell and Sejnowski on information processing in the V1, segmentation of auditory scenes in time-frequency space, and object detection from olfactory signals. In our expanded introductory discussion of ICA, we will carefully discuss these points using the references below [i]-[ii]-[iii] and contextualize [5].
>
> Now that we have clarified the setup in which our model operates, we believe that the model can be considered more biologically plausible in its simplicity. In particular, due to the stationary nature of the environment, the model would not re-learn entirely. The constraint is that the changes in the environment must happen slower than the speed of learning. As a result, we assume that synapses are updating fast but “slower” than the possible changes in the environment.
>
> Nonetheless, as per a point raised later by the reviewer, this is only a fraction of the task that the brain might be solving, and we are not considering other cognitive processes, top-down, such as attention mechanism or supervised learning. We will add to the introduction and the discussion that our algorithm is unsupervised and, as a result, is only a part of the solution to the puzzle. We will mention that significant efforts are currently undertaken to build biologically plausible reinforcement learning and supervised learning algorithms [v]-[vi]-[vii], which are more top-down approaches.
>
> While acknowledging that there is so much more to be understood about blind source separation in animal brains, we believe that we picked up where these early works left and made an important advance in the search for a biologically plausible multichannel ICA network.
>
> [i] Haykin, S., & Chen, Z. (2005). The cocktail party problem. Neural computation, 17(9), 1875-1902.
> [ii] McDermott, J. H. (2009). The cocktail party problem. Current Biology, 19(22), R1024-R1027.
> [iii] Hendin, O., Horn, D., & Hopfield, J. J. (1994). Decomposition of a mixture of signals in a model of the olfactory bulb. Proceedings of the National Academy of Sciences, 91(13), 5942-5946.
> [iv] Guerguiev, J., Lillicrap, T. P., & Richards, B. A. (2017). Towards deep learning with segregated dendrites. Elife, 6, e22901.
> [v] Sacramento, J., Ponte Costa, R., Bengio, Y., & Senn, W. (2018). Dendritic cortical microcircuits approximate the backpropagation algorithm. Advances in Neural Information Processing Systems, 31, 8721-8732.
> [vi] Payeur, A., Guerguiev, J., Zenke, F., Richards, B. A., & Naud, R. (2021). Burst-dependent synaptic plasticity can coordinate learning in hierarchical circuits. Nature neuroscience, 1-10.
>
>
>
> ## - Response to Q2:
>
> We will provide a more detailed description of what is currently known about primary sensory cortices and BSS. We mentioned above the work of Haykins and Chen [i] and McDermott [ii] that have clarified what was known in the early 2000s. Using the extra-page, we will elaborate on possible mappings to known neural structures. In section 5.3, lines 210-219, on neural mechanisms for the plasticity rules, we will elaborate on the mapping to the associated neural structures, inspired by the insightful work of [16].
>
> Regarding V1 mentioned by the reviewer, we believe that our algorithm is a possible candidate for V1 as it replicates some features of V1. We will elaborate on this point and show that our algorithm, trained on patches of images, recovers some localized and oriented filters that resemble Gabor filters, characteristic of V1 simple cells receptive fields. One of our current research directions is to include nonlinearities in neural activation functions, such as soft-thresholding, sigmoid, or ReLU, to force the model only to capture super-Gaussian distributions.
>
> Nonetheless, we will mention that there is no definitive answer if ICA-like models, as in [vii]-[viii] or sparse-coding-like models, best describe the processing performed in the early visual cortex. We will refer to the seminal work of Olshausen and Field [ix] and the more recent work of [x] for other sensory modalities. We will make it clear to the reader that it is still an ongoing discussion, and a large body of work exists in favor of both approaches.
>
> It is still critical to acknowledge that although primary visual, auditory, and olfactory cortices have been highly studied, it is still unknown what computational tasks and evident optimization problems they perform, and we are still speculating in good faith. We are also aware that the price paid for the clarity of the normative approach is that it does not reproduce every known biological observation. Our results highlight which experimental observations, such as two-compartment neurons, neuromodulatory signals, are essential for the circuit to implement ICA.
>
> [vii] Hyvärinen, Aapo, and Patrik O. Hoyer. "Topographic independent component analysis as a model of V1 organization and receptive fields." Neurocomputing 38 (2001): 1307-1315.
> [viii] Bartlett, M. S., & Sejnowski, T. J. (1997). Viewpoint invariant face recognition using independent component analysis and attractor networks. In Advances in neural information processing systems (pp. 817-823).
> [ix] Olshausen, B. A., & Field, D. J. (1997). Sparse coding with an overcomplete basis set: A strategy employed by V1?. Vision research, 37(23), 3311-3325.
> [x] Bhand, M., Mudur, R., Suresh, B., Saxe, A., & Ng, A. (2011). Unsupervised learning models of primary cortical receptive fields and receptive field plasticity. Advances in neural information processing systems, 24, 1971-1979.

---

> > ### Author Response · Authors · 2021-08-10
> > **Response to reviewer yveC - Response to Q3 and other suggestions**
> >
> > ## - Response to Q3
> >
> > Thank you for raising this point. Yes, we have evaluated our model against other ICA algorithms. Using the extra page and the appendix section, we will add these results to the final manuscript.
> >
> > - Evaluation Scenarios:
> >
> > We evaluated the models on four different scenarios.
> >
> > Scenario 1: Data are white and composed of 3 independent sub-Gaussian sources
> > Scenario 2: Data are colored and again composed of only 3 sub-Gaussian sources.
> > Scenario 3: Data are white and composed of 2 sub-Gaussian sources and 1 super-Gaussian source.
> > Scenario 4: Data are colored and composed of 2 sub-Gaussian sources and 1 super-Gaussian source.
> >
> > - Metric:
> >
> > To quantify the performance of the algorithms, we use the mean-squared error,
> > $$ error(t) = \frac{1}{td} \sum_{t=1}^T || x_t - Py_t ||^2; $$
> > With $t$ the number of samples and $d$ the dimension of the sources.
> >
> > - Competing algorithms:
> >
> > The algorithms that we evaluated against are the following:
> > The Herault Jutten (HJ) algorithm [xi]:
> > The unmixing matrix is sought in the form $W = ( I+ S^{-1})$ and the off-diagonal elements of S are updated using the rule:
> > $$ S_{t+1} = S_{t} + \eta g(y_t) h(y_t)^\top ~~, $$
> > with $h$ and $g$ two possibly different linear or nonlinear component-wise functions. For sub-Gaussian distributions we use $g(y) = y^3$ and $h(y) = y$.
> >
> > The EASI algorithm [xii]-[xiii] is closely related to a nonlinear PCA-type network.
> > $$ W_{t+1} = W_{t} - \eta ( y_t y_t^\top - I + g(y_t) h(y_t)^\top - h(y_t) g(y_t)^\top ) W_{t} ~~. $$
> > For sub-Gaussian distributions we use $g(y) = y$ and $h(y) = tang(y)$.
> >
> > The Bell and Sejnowski algorithm [xiv] from an information theoretic approach (Infomax):
> > $$ W_{t+1} = W_{t} + \eta ( W_{t}^{-1} + g(y_t) x_t^\top ) ~~. $$
> > We use $g(y) = -2 tanh(y)$.
> >
> > The Amari algorithm [xv], based on Infomax above.
> > $$ W_{t+1} = W_{t} + \eta ( I - 2 g(y_t) y_t^\top ) W_{t} $$
> > We use $g(y) = -2 tanh(y)$.
> >
> > And finally the nonlinear Oja algorithm [xvi]-[xvii]:
> > $$ W_{t+1} = W_{t} + \eta ( g(y_t) x_t^\top - g(y_t) g(y_t)^\top W_t^\top) $$
> > with again $g(y) = tanh(y)$.
> >
> > We chose these models because they were designed with neural architectures in mind and are seminal works on neural ICA algorithms. However, like Nonlinear Oja that we mentioned in Section 5.3.1, these models suffer from biological implausibility. We will describe their limitations in the appendix and leave it for the interested reader.
> >
> > - Results:
> >
> > In brief, our model outperforms others models on scenarios 2-3-4 and is competitive with the Amari algorithm, which is an improved version of Infomax and is known to be significantly faster. Naturally, in scenario 1, all models perform the task perfectly, with the Nonlinear Oja and Amari algorithm being the fastest.
> > Out of the other algorithms, EASI performs well on scenarios 1-2 but fails on tasks 3-4, where data are composed of both sub-Gaussian and super-Gaussian distributions. The other models are known to require pre-whitening and fail on scenarios 2 and 4, naturally.
> > We have prepared a log-log plot of convergence with error bars that illustrate the above mentioned results.
> >
> > The results show that our algorithm performs perfect separation with both whitened and colored data, and with sources both sub and super-Gaussian simultaneously.
> >
> > [xi] Jutten, Christian, and Jeanny Herault. "Blind separation of sources, part I: An adaptive algorithm based on neuromimetic architecture." Signal processing 24.1 (1991): 1-10.
> > [xii] Laheld, Beate, and Jean-François Cardoso. "Adaptive source separation with uniform performance." Proc. EUSIPCO. Vol. 1. 1994.
> > [xiii] Cardoso, J-F., and Beate H. Laheld. "Equivariant adaptive source separation." IEEE Transactions on signal processing 44.12 (1996): 3017-3030
> > [xiv] Bell, A. J., & Sejnowski, T. J. (1995). An information-maximization approach to blind separation and blind deconvolution. Neural computation, 7(6), 1129-1159.
> > [xv] Amari, Shun-ichi, Andrzej Cichocki, and Howard Hua Yang. "A new learning algorithm for blind signal separation." Advances in neural information processing systems. Morgan Kaufmann Publishers, 1996.
> > [xvi] Karhunen, Juha, Liuyue Wang, and Ricardo Vigario. "Nonlinear PCA type approaches for source separation and independent component analysis." Proceedings of ICNN'95-International Conference on Neural Networks. Vol. 2. IEEE, 1995.
> > [xvii] Oja, E. (1997). The nonlinear PCA learning rule in independent component analysis. Neurocomputing, 17(1), 25-45.
> >
> >
> > ## - Regarding the other suggestions from the reviewer.
> >
> > We will add a sentence mentioning that at equilibrium, the derivative on the left-hand side is equal to zero, which leads to the equation $y_t = M^{-1} c_t$, which is the same as the one presented in the offline setting but does not require inverting the $M$ matrix.
> >
> > The other two comments, regarding respectively the specificity of the algorithm and the top-down cognitive processes in the brain have been addressed along with Q1 and Q2. We can further elaborate on them if the reviewer wishes us to.

---

> > > ### Comment · Reviewer_yveC · 2021-08-19
> > > **Thank you for the clarifications / still not convinced regarding biological plausibility**
> > >
> > > Thank you, the responses answered most of my questions.
> > >
> > > The one issue I am still not convinced about is whether the proposed algorithm is biologically plausible.
> > >
> > > - Could the authors give a concrete example of a real-world problem that a biological organism may encounter and where the the brain might use the proposed ICA algorithm? On what timescale would the biological brain solve this problem (hundreds of milliseconds, seconds, minutes?), and what do the $x_t$ correspond to?
> > >
> > > - In my initial review, I asked whether I understood the following correctly:
> > > > In Algorithm 1 and Section 5.1, the authors explain that computing independent components involves updating the “weight matrix W encoded by the feedforward synapses connecting the input neurons to the output neurons”. This would imply that synapses re-learn completely new weights every time that the set of neurons computes independence components.
> > >
> > >   The authors replied:
> > > > ... In particular, due to the stationary nature of the environment, the model would not re-learn entirely.
> > >
> > >   I realize that my initial question was phrased imprecisely. I meant that synapses would need to be updated "significantly" (not "completely") for each input that is being processed. From Algorithm 1 in the paper, I understand that for a given input $x$, matrix $W$ gets updated in a loop until convergence on input $x$. When a different input $x'$ gets processed afterwards, the matrix $W$ again needs to be updated until convergence. In other words, $W$ is treated like working memory rather than as a set of parameters that gets learned to work for a large set of inputs. Is this understanding correct?

---

> > > > ### Author Response · Authors · 2021-08-20
> > > > **Response to reviewer yveC on biological plausibility and inner-workings of our algorithm**
> > > >
> > > > We are glad that our first response answered most of the questions you raised. We hope that the following will clear out the last reservations that you might have left.
> > > >
> > > > > Question 1: The one issue I am still not convinced about is whether the proposed algorithm is biologically plausible. Could the authors give a concrete example of a real-world problem that a biological organism may encounter and where the brain might use the proposed ICA algorithm? On what timescale would the biological brain solve this problem (hundreds of milliseconds, seconds, minutes?), and what do the xt correspond to?
> > > >
> > > > We had mentioned several brain areas that might involve an ICA-like mechanism, including the visual and the auditory cortex. Since we discussed Bell and Sejnowski's proposal for V1 receptor field formation [1]-[2], we will choose it as an example. The inputs $x_t$, namely patches of an image/visual input, can be viewed as a nearly independent combination of simpler features, and one can train neurons to recognize patterns like edges, as is well known to the community. There is a large body of experimental literature on the plasticity of this process, including experiments where a different statistic of stimulation is applied, resulting in the loss of orientation sensitivity, for example [3]. We believe these changes take many hours to several days. The statistics of natural images is stable enough so that even days-weeks of learning time is viable.
> > > >
> > > > While we might pick the visual system as an example, we hope that the ICA-like mechanism would not be limited to just V1 or the visual cortex. Such mechanisms may be at play in the auditory cortex as well for auditory feature extraction [4]. In fact, retinal projections, rerouted to the auditory cortex, lead the auditory cortex to form similar localized orientation sensitive neurons [5]. This suggests some general-purpose learning mechanisms could be active across the neocortex.
> > > >
> > > > Perhaps we do not fully understand where the reviewer's main objection is. Does the reviewer not believe that ICA is a good candidate for canonical processing in the brain across various stimuli modalities? Works such as [6,7,8,9,10] suggest that natural stimuli's underlying simplicity can often be expressed within this framework. An alternative possibility is that the reviewer thinks that ICA is a good candidate but is not utilized by the brain because of some issues with the timescale of learning? Experiments training cultured neurons to perform blind source separation [11] suggest that several hours should be enough, a number that is not inconsistent with some observations in live animals [12]-[13].
> > > >
> > > > In our paper, we also referred to the work of [14] that hypothesized that specific cerebellar glomeruli are configured so that the instantaneous external calcium concentration will encode the level of spike activity in postsynaptic cells. They concentrated on the specialized glomeruli in the cerebellum at the interface of the mossy fiber and granule cell layers and showed that such a model would perform a form of ICA.
> > > >
> > > >
> > > > > Question 2: I realize that my initial question was phrased imprecisely. I meant that synapses would need to be updated "significantly" (not "completely") for each input that is being processed. From Algorithm 1 in the paper, I understand that for a given input x, matrix W gets updated in a loop until convergence on input x. When a different input x′ gets processed afterwards, the matrix W again needs to be updated until convergence. In other words, W is treated like working memory rather than as a set of parameters that gets learned to work for a large set of inputs. Is this understanding correct?
> > > >
> > > > Thank you for your rephrased question. To be honest, we are confused by the phrasing "$W$ is treated like working memory **rather than** as a set of parameters." It is common in models of neural learning for parameters like $W$, representing synaptic weights, to be updated based on activity in the neural network, so we do not know what to make of the 'rather than' part.
> > > >
> > > > The learned parameters are "a set of parameters that get learned to work for a large set of inputs" and not a "working memory." Indeed after enough training steps, the parameters $W$ do capture something about the statistics of past ${x_t}$s. Using the reviewer's words, the synaptic weights will not change "significantly" after converging to the optimal solution. If future $x_t$s are chosen with the same statistical properties as past $x_t$s, W does not change, on average. The size of the stochastic fluctuations around the average value depends on the learning rate. Provided individual input causes a small change; the resulting fluctuation would also be small.
> > > >
> > > > To put it in perspective, the reviewer is probably familiar with the inner workings of Oja's algorithm [15]. Our model operates very similarly. It is well-known that, unlike in models of associative memory, e.g. Hopfield model [16], neural networks trained with Oja's rule will converge to the principal components. At every training step using Oja’s algorithm, the weights change, but after enough training, the weights align with the optimal solution and stop updating "significantly", they fluctuate around this optimal solution. Here, instead, the weights converge to the independent components instead of the principal components and fluctuate around this solution.
> > > >
> > > > To clarify the parallel with Oja's algorithm, we dissect both algorithms below. We add a training time index to $W$ to specify that $W_{t-1}$ is obtained after $t-1$ training steps, resp. $W_{t}$ is obtained after $t$ training.
> > > >
> > > > Let us assume that $t-1$ training steps have been performed, we are given the synaptic matrices, $W_{t-1},M_{t-1}$. Now we present a new input sample $x_t$.
> > > >
> > > > ### 1.  **Training of our algorithm at time $t$:**
> > > > The first step is to determine the output activity of the network, i.e., the $y_t$ associated with the $x_t$, using the current synaptic weights matrices.
> > > >
> > > > * Step 1: The neural dynamics below determines $y_t$ from $x_t$, $W_{t-1}$, and $M_{t-1}$ as:
> > > > $$ \frac{dy(\tau)}{d\tau} = W_{t-1} x_t - M_{t-1} y_t (\tau).  $$
> > > >
> > > > Now that we have obtained the $y_t$ associated with $x_t$ we update the weights $W_{t-1} $ and $M_{t-1}$ before the next sample is presented as,
> > > >
> > > > * Step 2a: feedforward connection update using $y_t, x_t$ and $W_{t-1}$  as
> > > > $$W_{t} = W_{t-1} + 2\eta(y_t - || y_t ||^2 \Lambda^2 - 2 W_{t-1}  x_t) x_t^\top $$
> > > >
> > > > * Step 2b: lateral connection update using $y_t$  as
> > > > $$ M_{t} = M_{t-1} +\frac{\eta}{\tau} (y_t y_t^\top  - I_d ) $$
> > > >
> > > > After training, once $W$ and $M$ stabilize, we have $​​W_t \approx W_{t-1}$. This is due to the fact that $(y_t - || y_t ||^2 \Lambda^2- 2W_{t-1}x_t)x_t^T$ is zero on the average.
> > > >
> > > > ### 2. **Training of Oja’s algorithm at time $t$:**
> > > > We now recall Oja’s algorithm. Oja’s algorithm is also a two-step algorithm like ours, and for example at time $t$ the training is performed as
> > > > * Step 1: neural dynamics
> > > > $$ \frac{dy(\tau)}{d\tau} = W_{t-1}^\top x_t - W_{t-1}^\top W_{t-1} y_t^\top (\tau).  $$
> > > >
> > > > * Step 2: update rule
> > > > $$ W_{t} = W_{t-1} + \eta ( x_t - W_{t-1} y_t) y_t^\top $$
> > > >
> > > > After training, once $W$ stabilizes, $​​W_t \approx W_{t-1}$. This is due to the fact that  $( x_t - W_{t-1} y_t) y_t^\top$ is zero on the average.
> > > >
> > > > We believe that the review paper from [17] could help clarify the similarities and differences between similarity-preserving algorithms and Oja’s learning rules. We also discussed a comparison between our model and Oja’s algorithm in Section 5.3.1.
> > > >
> > > >
> > > > [1] Bell, A., & Sejnowski, T. J. (1996). Edges are the 'Independent Components' of Natural Scenes. NeurIPS, 9, 831-837.
> > > >
> > > > [2] Bell, Anthony J., & Terrence J. Sejnowski. "The “independent components” of natural scenes are edge filters." Vision Research 37.23 (1997): 3327-3338.
> > > >
> > > > [3] Weliky, M., & Katz, L. C. (1997). Disruption of orientation tuning visual cortex by artificially correlated neuronal activity. Nature, 386(6626), 680-685.
> > > >
> > > > [4] Lewicki, M. S. (2002). Efficient coding of natural sounds. Nature Neuroscience, 5(4), 356-363.
> > > >
> > > > [5] Sharma, J., et al. (2000). Induction of visual orientation modules in auditory cortex. Nature, 404(6780), 841-847.
> > > >
> > > > [6] Barlow, H. B. (1989). Unsupervised learning. Neural computation, 1(3), 295-311.
> > > >
> > > > [7] Atick, J. J. (1992). Could information theory provide an ecological theory of sensory processing?. Network: Computation in neural systems, 3(2), 213-251.
> > > >
> > > > [8] Barlow H (2001) Redundancy reduction revisited. Network 12: 241–253.
> > > >
> > > > [9] Simoncelli E, Olshausen B (2001) Natural image statistics and neural representations. Annu Rev Neuroscience 24: 1193–1216.
> > > >
> > > > [10] Simoncelli E (2003) Vision and the statistics of the visual environment. Curr Op Neurobiol 13: 144–149.
> > > >
> > > > [11] Isomura, Takuya, et al. "Cultured cortical neurons can perform blind source separation according to the free-energy principle." PLoS CompBio 11.12 (2015): e1004643.
> > > >
> > > > [12] Weliky, M. (1999). Recording and manipulating the in vivo correlational structure of neuronal activity during visual cortical development. Journal of neurobiology, 41(1), 25-32.
> > > >
> > > > [13] Espinosa, J. S., & Stryker, M. P. (2012). Development and plasticity of the primary visual cortex. Neuron, 75(2), 230-249.
> > > >
> > > > [14] Coenen, O. J. M. D., & Sejnowski, T. J. (2001). Cerebellar glomeruli: Does limited extracellular calcium direct a new kind of plasticity. In Society for Neuroscience Abstracts (Vol. 27).
> > > >
> > > >  [15] Oja, E. (1989). Neural networks, principal components, and subspaces. International journal of neural systems, 1(01), 61-68.
> > > >
> > > > [16] Hopfield, J. J. (1982) Neural networks and physical systems with emergent collective computational properties. Proc. Nat. Acad. Sci. (USA) 79, 2554-2558.
> > > >
> > > > [17] Pehlevan, C., & Chklovskii, D. B. (2019). Neuroscience-inspired online unsupervised learning algorithms: Artificial neural networks. IEEE Signal Processing Magazine, 36(6), 88-96.

---

> > > > > ### Comment · Reviewer_yveC · 2021-09-08
> > > > > **I am sorry for the confusion**
> > > > >
> > > > > Dear authors, I updated my review and score after your most recent response. I had misunderstood the semantics of $x_t$. I apologize for that error.

---

### Official Review · Reviewer_ocjn · 2021-07-18

**Rating:** 8
**Confidence:** 3

**Summary:**

The authors propose a compartmental single-layer neural network model to be used in an online ICA algorithm utilizing a local synaptic update rule. The novel algorithm is based on the FOBI procedure for PCA by restating the objective function using similarity matching. As opposed to hebbian update rules, the synaptic updates used in the present paper incorporate a time-varying modulating factor which is computed as a function of the total output activity. Biological plausibility of this relaxed locallity constraint is justified by hinting at experimential results showing the importance of a third factor in the outcome of hebbian plasticity. The approach is verified using three different experiments including synthetic and real-world data.

**Limitations And Societal Impact:**

Limitations and possible remedies are listed. Due to the specific mathematical nature of the the paper its societal impact is unlikely to exceed the impact of the field as a whole.

**Main Review:**

Originality:
The search for a local learning rule implementing ICA has a long history and many approaches have been investigated in the past. The presented approach is comparable with the work of Isomura and Toyoizumi which is also noted in the paper. While the mathematical details elude the reviewer, the implications and resulting learning rules seem quite similar. The main difference appears to be trading off a constant "error baseline" for lateral inhibitory connections. A more thourough comparison of the two approaches could further improve the quality of the paper.

Quality:
The submission seems to be technically sound. Claims are supported by mathematical proofs and adequate experimental results. The authors are honestly stating the weaknesses of their approach.

Clarity:
The approach is laid out in a clear and understandable fashion. Included figures are helpful for understanding the resulting NN structure and the correspondence to the FOBI procedure.

Significance:
Introducing yet another biologically plausible method for ICA might be considered of mediocre importance. Nonetheless, the work may advance our understanding of the inner workings of biological neural networks by linking a novel objective function to a simple network motif.


**Time Spent Reviewing:**

8

---

> ### Author Response · Authors · 2021-08-10
> **Response to reviewer ocjn**
>
> We thank reviewer ocjn for their positive review and are responding to the questions raised.  Below we address the comments and will incorporate all feedback in the final version.
>
> - Comparison to EGHR:
>
> We will provide a detailed discussion of the contrasts between our work and the EGHR algorithm of Isomura and Toyozumi. As the reviewer mentions, their approach avoids using pairwise inhibitory interaction, something that is crucial for our work. Our objective function, when sources have distinct distributions, has a global optimum. Numerical and theoretical analysis of the performance of the EGHR  algorithm relies on source distributions being the same and has several equivalent optima. Our model, instead, is best suited for separating mixtures of sources, both sub and super-Gaussian. We have performed numerical evaluations of the convergence of our model on different scenarios and will add them to the final manuscript.
>
> Since EGHR depends upon an unspecified a priori energy function, if the referee has a specific choice in mind, we could explore how it functions, especially when we mix sub- and super-Gaussian sources.

---

### Author Response · Authors · 2021-08-10
**General response to the reviewers**


We thank the reviewers for their thoughtful and positive feedback. We are encouraged that they found our submission clear [h9nv, QCCE, up9Q, EXE1, yveC, ojcn], novel, original [h9nv, QCCE, up9Q, EXE1, yveC, ojcn]. We are also encouraged that the reviewers find it more biologically plausible than previous work [up9Q] and having the potential to advance our understanding of the inner workings of biological neural networks [ojcn]. We are also pleased that the reviewers appreciate the normative approach followed to derive the bio-plausible learning rules and the resulting interpretation of the neural mechanisms [QCCE, up9Q, h9nv].

We appreciate the overall positive reaction of the reviewers and the constructive feedback, which we will address individually. In response to the feedback, we performed several extra supporting experiments that we will include in the revised manuscript.

---

### Decision · Program_Chairs · 2021-09-27

**Decision:**

Accept (Spotlight)

**Comment:**

Dear authors,

reviewers have reached a clear positive consensus on your work and I
therefore endorse your paper for acceptance.

Best regards
The AC